# Myelin plasticity in the ventral tegmental area is required for opioid reward

Belgin Yalçın[1], Matthew B. Pomrenze[2], Karen Malacon[1], Richard Drexler[1], Abigail E. Rogers[1], Kiarash Shamardani[1], Isabelle J. Chau[1], Kathryn R. Taylor[1], Lijun Ni[1], Daniel Contreras-Esquivel[1], Robert C. Malenka[2] & Michelle Monje[1,3 ✉]

All drugs of abuse induce long-lasting changes in synaptic transmission and neural circuit function that underlie substance-use disorders[1,2]. Another recently appreciated mechanism of neural circuit plasticity is mediated through activity-regulated changes in myelin that can tune circuit function and influence cognitive behaviour[3–7]. Here we explore the role of myelin plasticity in dopaminergic circuitry and reward learning. We demonstrate that dopaminergic neuronal activity-regulated myelin plasticity is a key modulator of dopaminergic circuit function and opioid reward. Oligodendroglial lineage cells respond to dopaminergic neuronal activity evoked by optogenetic stimulation of dopaminergic neurons, optogenetic inhibition of GABAergic neurons, or administration of morphine. These oligodendroglial changes are evident selectively within the ventral tegmental area but not along the axonal projections in the medial forebrain bundle nor within the target nucleus accumbens. Genetic blockade of oligodendrogenesis dampens dopamine release dynamics in nucleus accumbens and impairs behavioural conditioning to morphine. Taken together, these findings underscore a critical role for oligodendrogenesis in reward learning and identify dopaminergic neuronal activity-regulated myelin plasticity as an important circuit modification that is required for opioid reward.

Motivated behaviour—critical for adapting to the environment and therefore for animal survival—depends on proper function of reward circuitry. Drugs of abuse induce maladaptive and persistent modifications in reward circuitry, facilitating the development of addictive behaviours. All drugs of abuse, including opioids, target the midbrain dopaminergic (DA) reward system and induce lasting changes in synaptic transmission and neural circuit function[1,2]. Morphine, a natural opioid found in opium, triggers synaptic plasticity and alters neuronal function in the ventral tegmental area (VTA) and nucleus accumbens (NAc), two key structures of the DA reward system promoting a pathological form of reward learning[1,2]. These experience-dependent alterations of reward system function are critical for shaping drug-induced behavioural changes and hence the development of substance-use disorders[1,2].

Although a significant role for microglia and astrocytes in these neural circuit modifications is becoming increasingly apparent[8,9], contributions of oligodendroglial lineage cells and myelin to reward circuit adaptation in health and maladaptation in addictive states remain unknown. However, oligodendroglial lineage cells are a glial cell type particularly well positioned to contribute to structural and functional changes in addiction-related neural circuits. Oligodendrocytes generate myelin which ensheaths axons to modulate conduction velocity[10] and provide metabolic support to axons[11,12], therefore playing a fundamental role in shaping neural transmission. Neuronal activity can regulate myelination during development[13,14]

and myelin plasticity in adulthood[3,15–20]. Activity-regulated myelin changes can occur by means of oligodendrocyte precursor cell (OPC) proliferation, oligodendrogenesis and de novo myelination or through myelin remodelling by existing oligodendrocytes[3,16–19,21]. Even small changes in myelination can tune circuit dynamics[22–24] and consequently influence cognition and behaviour[3–7,25,26]. These circuit-specific, activity-regulated myelin changes seem to occur only in distinct neuronal types[3,18]. Which neuronal subtypes, in which circuits and with which patterns of activity elicit oligodendroglial responses remain to be fully understood. Furthermore, much remains to be learned about how activity-regulated myelination may become maladaptive and contribute to the pathophysiology of neurological[27] or psychiatric[28] diseases.

Here we explore the role of myelin plasticity in morphine-elicited reward and dopamine release dynamics in the NAc. Our results identify myelin plasticity as a previously unappreciated feature of the activity-dependent modifications of reward circuit function which critically contribute to the behavioural reinforcing effects of opioids.

## DA neuron activity regulates myelination

We leveraged in vivo optogenetic strategies to examine whether DA neuron activity regulates oligodendroglial cell dynamics in reward circuitry. To directly stimulate DA neurons, we injected adeno-associated virus (AAV) carrying Cre-inducible channelrhodopsin2 (ChR2) and

[1]Department of Neurology and Neurological Sciences, Stanford University, Stanford, CA, USA. [2]Nancy Pritzker Laboratory, Department of Psychiatry and Behavioral Sciences, Stanford University, Stanford, CA, USA. [3]Howard Hughes Medical Institute, Stanford, CA, USA. ✉e-mail: mmonje@stanford.edu

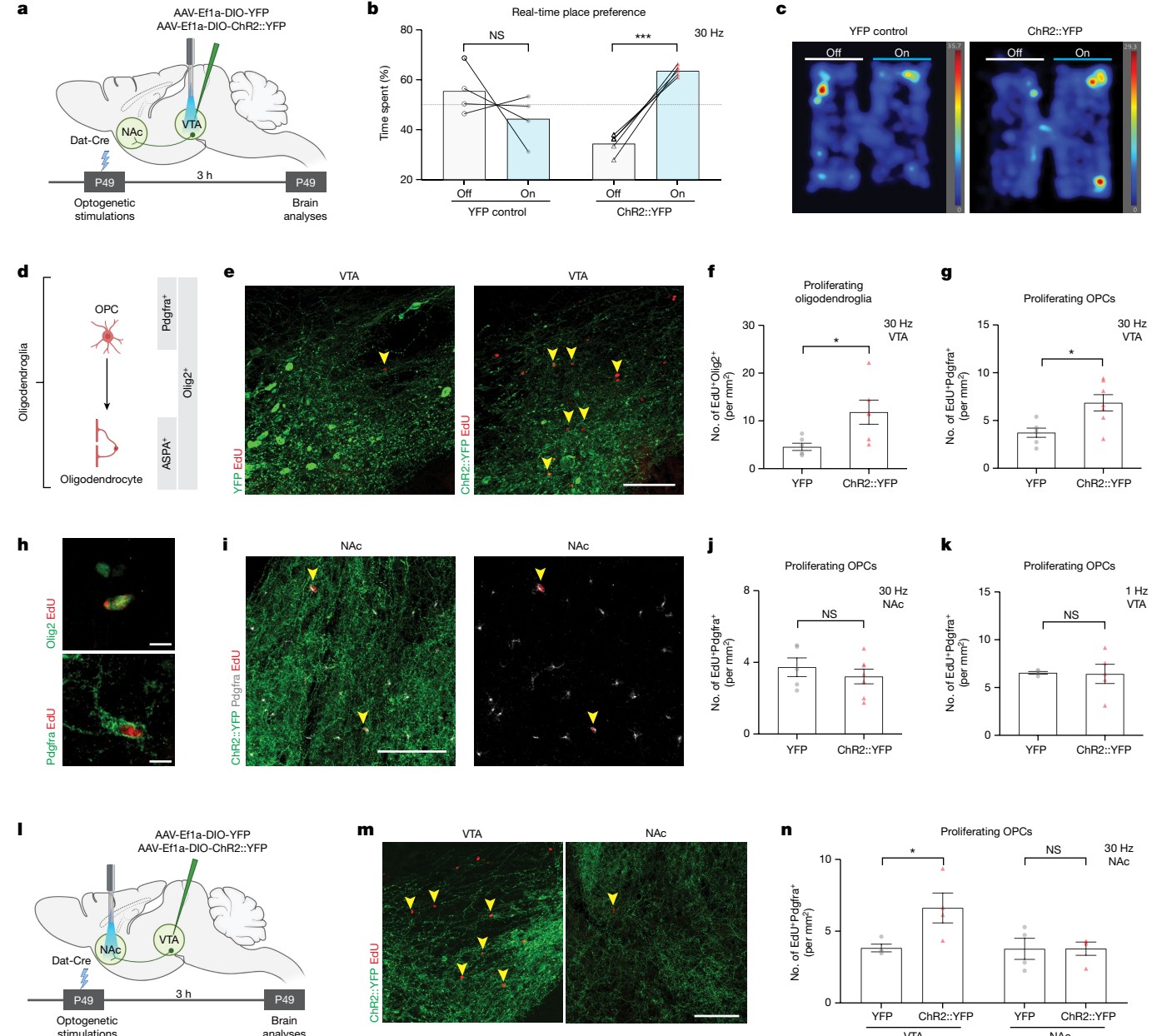

**Fig. 1 | Reward circuitry DA neuron activity increases OPC proliferation in VTA. a**, Experimental paradigm for DA neuron stimulation in VTA. **b**, The 30 Hz optogenetic stimulation of DA neurons increases real-time place preference (*n* = 4 YFP control mice; *n* = 5 ChR2::YFP mice). **c**, Representative heatmaps illustrating the time spent in different compartments during the real-time place preference test. **d**, Schematic of oligodendroglial lineage; Olig2 is an oligodendroglial lineage marker and Pdgfrα marks OPCs and ASPA marks mature oligodendrocytes. **e–k**, Optogenetic stimulation of DA neurons (YFP, green) results in oligodendroglial precursor proliferation in VTA. Confocal micrographs of proliferating cells (EdU⁺, red) in VTA (**e**) and in NAc (**i**). Scale bars, 100 μm. The 30 Hz stimulation of VTA DA neurons increases proliferative oligodendroglial lineage (Olig2⁺EdU⁺) (**f**) and oligodendrocyte precursor

(Pdgfrα⁺EdU⁺) (**g**) cells in VTA (*n* = 6 mice per group). **h,i**, Representative confocal images. Scale bars, 10 μm. **j,k**, OPC proliferation (Pdgfrα⁺EdU⁺) does not change in NAc after 30 Hz VTA stimulation (**j**) or in VTA after 1 Hz VTA stimulation (**k**) (*n* = 4 YFP control mice; *n* = 5 ChR2::YFP mice). **l**, Experimental paradigm for optogenetic stimulation of DA axons (YFP, green) in NAc. **m**, Proliferating cells (EdU⁺, red, arrowheads) in NAc and VTA after 30 Hz NAc stimulation of DA axons (ChR2::YFP, green). Scale bars, 100 μm. **n**, The 30 Hz NAc stimulation of DA axons increases OPC proliferation (Pdgfrα⁺EdU⁺) in VTA but not in NAc (YFP and ChR2::YFP, *n* = 4). Scale bars, 100 μm. **b**, Paired two-tailed *t*-test for each condition. **h,l,j,k**, Unpaired two-tailed *t*-test. NS, not significant *P* > 0.5, *\*P* < 0.05, *\*\*P* < 0.01, *\*\*\*P* < 0.001. Each data point is one mouse; data shown as mean; error bars, s.e.m. **a,d,l**, Schematics created with BioRender.com.

YFP or YFP alone into the VTA of Dat-Cre mice (Fig. 1a and Extended Data Fig. 1a). This method achieves DA neuron-specific expression of ChR2 or YFP in VTA, where DA neuron cell bodies are localized and also in their axonal projections towards other brain regions including the main downstream target, NAc. Localization of ChR2 and YFP to DA neurons was confirmed by tyrosine hydroxylase (TH) colocalization in VTA (Extended Data Fig. 1b). Optogenetic stimulation of VTA at 30 Hz

increased cfos in ChR2⁺ DA neurons, indicating successful optogenetic targeting (Extended Data Fig. 1c–e).

DA neurons exhibit different activity modes; 'phasic firing' indicates synchronized bursts of action potentials at frequencies above 10 Hz, whereas 'tonic firing' refers to spontaneous activity between 0.2 and 10 Hz (refs. 29,30). Because tonic and phasic firing can mediate different aspects of motivated behaviour, we tested both activity modes. With

stimulation of VTA at 30 Hz, ChR2-expressing mice showed a strong preference for the stimulation-paired chamber in a real-time place preference assay, confirming a rewarding effect of phasic DA neuron activation[30] (Fig. 1b,c). To assess whether this reinforcing phasic activity affects oligodendroglial lineage cells (Fig. 1a), we used oligodendroglial cell identity markers Olig2 (all oligodendroglial cells), Pdgfrα (OPCs) and ASPA (mature oligodendrocytes; Fig. 1d). To mark dividing cells, we administered the thymidine analogue 5-ethynyl-2′-deoxyuridine (EdU) before each optogenetic stimulation session. Phasic stimulation of VTA DA neurons at 30 Hz intermittently over 30 min increased proliferation of oligodendroglial precursor cells (EdU$^+$Olig2$^+$ and EdU$^+$Pdgfrα$^+$ colabelled cells) in VTA within 3 h (Fig. 1e–h). Notably, there was no increased oligodendroglial precursor cell proliferation in the NAc, where axons and presynaptic terminals of DA neurons are located (Extended Data Fig. 2a,b and Fig. 1i–j). By contrast, tonic stimulation, which does not cause real-time reward or aversion[30], at 1 Hz for 30 min did not change oligodendroglial precursor cell proliferation in VTA, suggesting that the firing mode of DA neurons is critical for the activity-regulated oligodendroglial response (Extended Data Fig. 2a,c and Fig. 1k).

As we observed a proliferative increase only in VTA OPCs (Fig. 1g), we next investigated whether the oligodendroglial response to DA neuron activity is brain-region specific. We once again expressed ChR2 in VTA DA neurons but placed the fibre over the NAc and stimulated at 30 Hz for 30 min as above (Fig. 1l). Phasic stimulation of VTA DA axon terminals in the NAc increased proliferating oligodendroglia (EdU$^+$Olig2$^+$) and OPCs (EdU$^+$Pdgfrα$^+$) in the VTA (Extended Data Fig. 2d,e and Fig. 1m,n) probably because of antidromic activation of VTA DA neuron cell bodies[31]. However, it did not change proliferation of oligodendroglial precursor cells locally in NAc (Extended Data Fig. 2d,e and Fig. 1m,n). GABAergic inhibitory neurons in VTA regulate the activity of DA neurons[32]. To test the role of elevated DA neuronal activity using an alternative strategy that does not directly manipulate DA neurons, we increased DA activity by inhibiting the GABAergic neurons in the VTA. We injected AAV carrying Cre-inducible halorhodopsin or YFP into the VTA of Vgat-Cre mice and optogenetically inhibited GABAergic neurons for 30 min (Extended Data Fig. 2f). Concordant with the findings described above, this inhibition of GABAergic neurons increased oligodendroglial (EdU$^+$Olig2$^+$) and OPC proliferation (EdU$^+$Pdgfrα$^+$) in VTA within 3 h (Extended Data Fig. 2f–h). Taken together, these data demonstrate that phasic VTA DA neuron activity promotes OPC proliferation specifically in VTA.

Next, we evaluated the cell fate of the new oligodendroglial cells proliferating in response to phasic DA neuron activity. To achieve a more sustained increase in neuronal activity, we stimulated DA neurons at 30 Hz for 10 min a day for 7 days and analysed the brains 4 weeks later (Fig. 2a). We examined new oligodendroglial lineage cells across the VTA→NAc pathway, including in the NAc and along the medial forebrain bundle (MFB), through which DA axons project. There was a substantial increase in oligodendrocyte (EdU$^+$ASPA$^+$) density in the VTA (Fig. 2b,e). However, neither NAc (Fig. 2c) nor MFB (Fig. 2d) showed a change in new oligodendrocytes (EdU$^+$ASPA$^+$) or new oligodendroglia (EdU$^+$Olig2$^+$; Extended Data Fig. 2i–k) even after a prolonged increase in DA neuron activity. Once again, these findings highlight the specificity for the activity-regulated oligodendroglial response in the VTA. This specificity may reflect the regional heterogeneity of OPCs across the brain[33] and/or the microenvironmental differences, such as proximity to DA cell bodies versus axons[34], or may reflect neuron subtype-specific biology[3,18].

As a mature oligodendrocyte can generate between 20 and 50 myelin sheaths, even a few oligodendrocytes can exert significant effects on myelination and conduction velocity[15,24]. To assess potential myelination changes in VTA as a result of activity-regulated oligodendrogenesis, we measured myelin basic protein (MBP) over ChR2$^+$ and YFP$^+$ areas. After daily phasic DA neuron stimulation, MBP intensity was significantly increased in the VTA of ChR2::YFP animals compared with those expressing YFP control (Fig. 2f,g), implying a change in myelination.

Next, we performed electron microscopy in VTA to delineate the myelination changes in this region (Fig. 2h). In the VTA of stimulated mice, we found increased myelinated axon density for medium-sized (500–1,000 nm) axons (Fig. 2i,j). Moreover, these medium-sized axons also exhibited thicker myelin sheaths, evident as a decrease in g-ratio (ratio of axon to myelinated axon diameters; Fig. 2k, l). To identify axons that exhibit myelin changes, we used immunogold labelling against YFP to detect DA axons. Of these medium-sized axons, 86% ± 4.63% were positive for immunogold labelling, indicating that the observed DA neuronal activity-regulated increase in myelination occurs on DA axons (Extended Data Fig. 2l,m).

## Morphine promotes VTA oligodendrogenesis

Drugs of abuse initiate persistent adaptations in reward circuitry, laying foundations for drug-seeking behaviours. Although every drug of abuse targets reward circuitry and alters DA signalling, each exerts effects through different mechanisms. We investigated whether acute administration of morphine or cocaine—two different drugs of abuse which work through distinct mechanisms—alters oligodendrogenesis as a part of these initial neural circuit adaptations. Wild-type mice were injected systemically with one dose of morphine (10 mg kg$^{-1}$, intraperitoneally), cocaine (15 mg kg$^{-1}$, intraperitoneally) or saline control and with EdU to mark the dividing cells (Fig. 3a). After 3 h, animals exposed to either morphine or cocaine exhibited an increase in the number of proliferating VTA OPCs (EdU$^+$Pdgfrα$^+$) compared to saline-treated controls (Fig. 3b,c). Similar to the oligodendroglial response to acute phasic optogenetic stimulation of DA neurons (Fig. 1g,n), this OPC increase was specific to the VTA (Fig. 3c).

Drugs of abuse recruit different components of reward circuits to adapt and mediate reward learning behaviour. We next investigated oligodendroglial cells after morphine- or cocaine-induced conditioned place preference (CPP), a form of Pavlovian conditioning in which a neutral context or cue becomes associated with the rewarding or aversive properties of a stimulus[35] (Fig. 3d). Wild-type mice were conditioned with morphine (10 mg kg$^{-1}$, intraperitoneally; or 20 mg kg$^{-1}$, intraperitoneally), cocaine (15 mg kg$^{-1}$, intraperitoneally) or saline for 5 days and administered EdU during conditioning sessions to concurrently trace proliferating OPCs. As expected, morphine- and cocaine-treated mice exhibited increasing locomotor activity in response to repeated injections, a form of behavioural sensitization[36] (Fig. 3e). They also exhibited a preference for the drug-paired conditioning chamber, that is CPP (Fig. 3f,g). These behavioural changes were accompanied by a substantial increase in the number of proliferating VTA OPCs (EdU$^+$Pdgfrα$^+$) for every drug-treated group (Fig. 3h). Neither morphine (10 mg kg$^{-1}$) nor cocaine evoked an increase in proliferating OPCs in NAc. However, morphine did cause an increase in NAc OPC proliferation at a higher (20 mg kg$^{-1}$) dose, suggesting that the same drug can recruit more OPC populations at higher doses (Fig. 3h). Concordantly, differentiation of these newly proliferated, EdU-marked OPCs in VTA into new oligodendrocytes (EdU$^+$ASPA$^+$) was evident by the end of this experimental paradigm (Fig. 3i,j).

To follow the changes in oligodendrocytes in VTA after morphine, we conditionally expressed mGFP in oligodendrocytes using a genetic strategy (mT-mG$^{lox/lox}$; Plp1-Cre$^{ERT}$) which sparsely labels differentiating oligodendrocytes at a timepoint 3 weeks before starting morphine (10 mg kg$^{-1}$) conditioning (Extended Data Fig. 3a). Morphine-treated animals exhibited increased locomotor activity across conditioning days and acquired a strong preference for the morphine-paired chamber (Extended Data Fig. 3b–d). Concordant with the EM findings described above, mGFP-labelled oligodendrocytes were found juxtaposed to TH$^+$ DA axons in VTA and wrapped around proximal segments of DA axons. Each oligodendrocyte in this region seemed to interact with several DA axons and exhibit ramified morphology (Extended Data Fig. 3e). Morphine increased the number of mGFP$^+$ oligodendrocytes

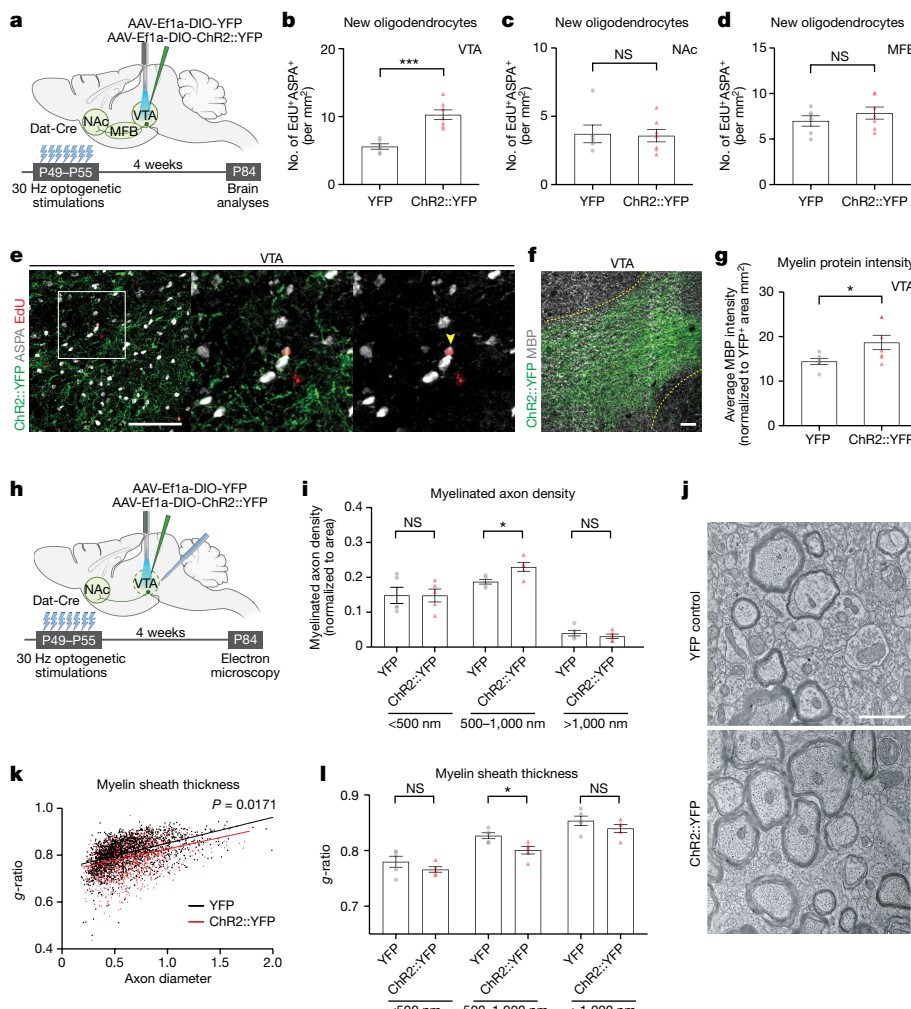

**Fig. 2 | DA neuron activity regulates myelination in VTA. a**, Experimental paradigm for optogenetic DA neuron stimulation (30 Hz) in VTA daily for 1 week, with brain analyses 4 weeks after the end of optogenetic stimulations. **b**–**d**, DA neuron stimulation increases oligodendrocytes (ASPA⁺ EdU⁺) in VTA (**b**) but not in NAc (**c**) or MFB (**d**) (n = 6 YFP control mice; n = 7 ChR2::YFP mice). **e**, Representative confocal micrographs of new oligodendrocytes (ASPA⁺ (grey), EdU⁺ (red, yellow arrowheads) in VTA. Scale bar, 50 μm. **f**, Myelination in VTA, yellow dotted line marks VTA border, DA neurons expressing ChR2::YFP (green); MBP (grey). Scale bar, 50 μm. **g**, DA neuron stimulation increases normalized myelin protein expression (ratio of average MBP intensity to DA neuron per axon area (mm²) in VTA) (n = 6 YFP control mice; n = 7 ChR2::YFP mice).

**h**, Experimental paradigm for electron microscopy analyses, as in **a**. **i**, DA neuron stimulation increases myelinated axon density of medium-sized (500–1,000 nm) axons. **j**, Electron micrographs show myelinated axons in VTA. Scale bar, 1 μm (n = 5 mice per group). **k**, Scatter plot of g-ratio as a function of axon calibre; YFP control axons (black dots) ChR2::YFP axons (red triangles). **l**, DA neuron stimulation reduces g-ratio of the medium-sized (500–1,000 nm) axons in VTA, indicating an increase in myelin sheath thickness. Unpaired two-tailed t-test. NS, not significant P > 0.5, *P < 0.05, ***P < 0.001. Each data point represents a mouse; data shown as mean; error bars, s.e.m. **a,h**, Schematics created with BioRender.com.

wrapped around TH⁺ axons and increased the area of mGFP⁺ oligodendrocyte processes in VTA (Extended Data Fig. 3f–h).

To understand whether this oligodendroglial response is specific to reward circuitry, we analysed proliferating OPCs in the premotor cortex. We found no change in the number of proliferating cortical OPCs across the drug-treated groups, suggesting that morphine and cocaine do not exert direct effects on OPCs globally but rather selectively influence OPCs in DA reward circuitry (Extended Data Fig. 4a–c).

As the oligodendroglial response to morphine is selectively found in VTA, we performed single-nucleus RNA sequencing (snRNA-seq) in the VTA after morphine conditioning (or saline control) to delineate interactions between oligodendroglia and other cellular populations (Extended Data Fig. 5a). We identified VTA cell types including various neuronal and glial types (Extended Data Fig. 5b) and the previously defined oligodendroglial subpopulations[37] (Extended Data Fig. 5c). CellChat analysis[38] showed interactions between neuronal and glial cell populations, demonstrating robust interactions between DA neurons

and OPCs (Fig. 3k). These findings strengthen the hypothesis that DA neurons can regulate oligodendrogenesis through their direct interactions with OPCs. Indeed, we found that morphine caused a shift in oligodendroglial subpopulations, causing transcriptional changes that are in line with the oligodendroglial differentiation required for oligodendrogenesis. For example, with morphine exposure, oligodendroglial cluster 4 shifted towards oligodendroglial cluster 9 by upregulating genes involved in lamellipodium organization, cell projection assembly and Wnt signalling which suggests a more differentiated oligodendroglial cell state (Extended Data Fig. 5d–f).

Oligodendroglial cells can express κ-opioid receptors[39,40] and may respond to the neuropeptide dynorphin in some contexts[41]. We therefore tested whether morphine or dopamine directly affects OPC dynamics in vitro and whether κ-opioid receptors mediate any direct effects of morphine. Incubation of OPCs with various concentrations of morphine did not change OPC proliferation (EdU⁺Pdgfrα⁺), in contrast to proliferative media positive control conditions, suggesting that the

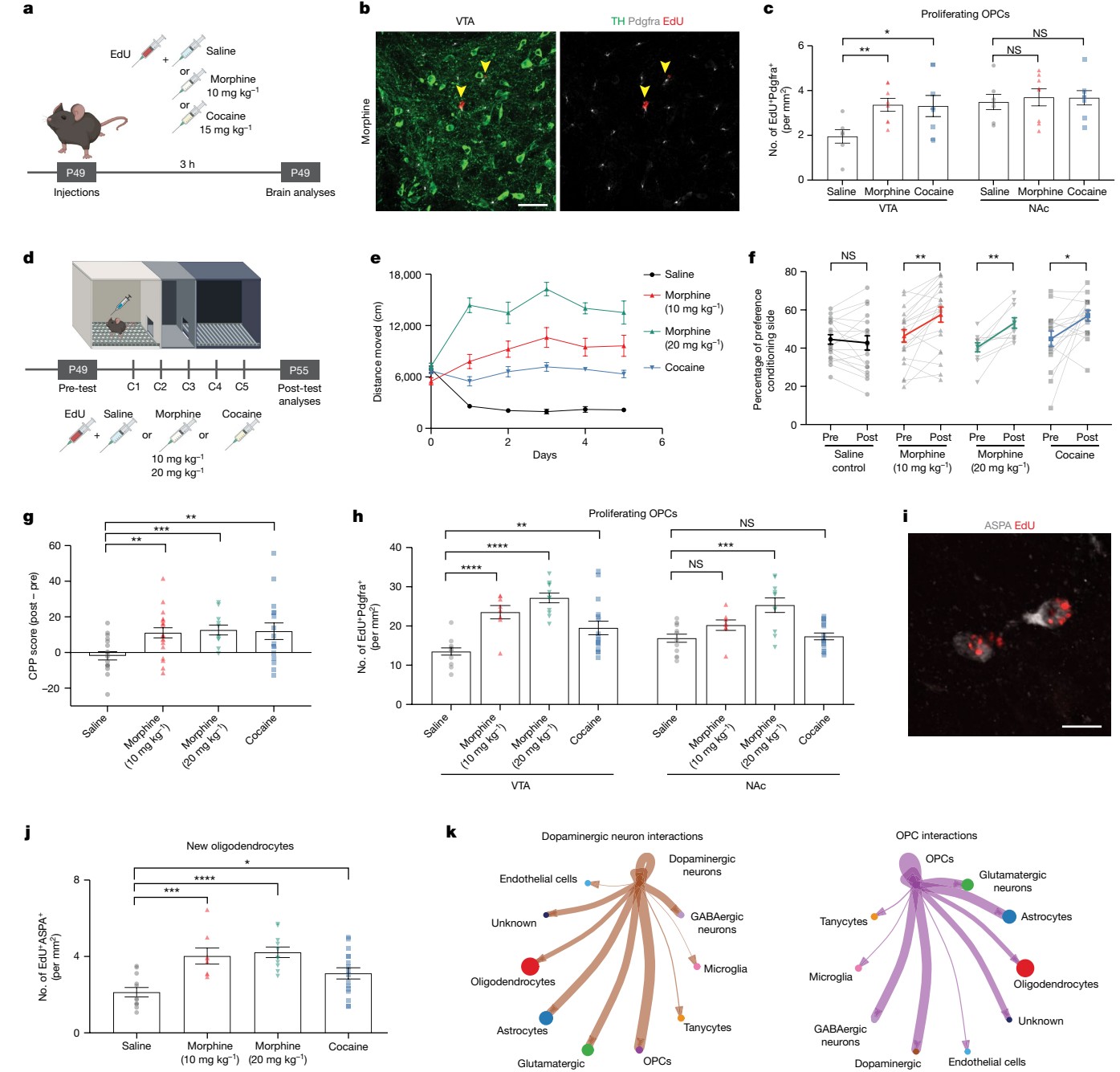

**Fig. 3 | Morphine and cocaine promote oligodendrogenesis in VTA.**
**a**, Experimental paradigm, OPC proliferation 3 h after administration of morphine or cocaine. **b**, Representative confocal micrograph of VTA, TH (DA neurons, green), Pdgfrα⁺ (OPCs, grey) and EdU⁺ (red). Scale bar, 50 µm. **c**, Single dose of morphine or cocaine increases proliferating OPCs (Pdgfrα⁺EdU⁺) in VTA but not in NAc (saline, $n = 7$ mice; morphine, $n = 8$ mice; cocaine, $n = 8$ mice). **d**, Experimental paradigm for CPP. **e**,**f**, Morphine or cocaine conditioning increases locomotor sensitivity (saline, $n = 12$ mice; morphine (10 mg kg⁻¹), $n = 12$ mice; morphine (20 mg kg⁻¹), $n = 11$ mice; cocaine, $n = 8$ mice) (**e**) and induces a robust place preference for the conditioning chamber (**f**). **g**, Morphine or cocaine conditioning increases CPP score (post-test − pre-test preference) (saline, $n = 18$ mice; morphine (10 mg kg⁻¹), $n = 20$ mice; morphine (20 m kg⁻¹), $n = 11$ mice; cocaine, $n = 16$ mice). **h**, Both morphine and cocaine conditioning

increase proliferative OPCs (Pdgfrα⁺EdU⁺) in VTA but only morphine at 20 mg kg⁻¹ increases OPC proliferation in NAc (saline, $n = 13$ mice; morphine (10 mg kg⁻¹), $n = 8$ mice; morphine (20 mg kg⁻¹), $n = 11$ mice; cocaine, $n = 16$ mice). **i**, New oligodendrocytes after morphine (10 mg kg⁻¹) exposure (ASPA⁺ EdU⁺). Scale bar, 10 µm. **j**, Both morphine and cocaine conditioning increase oligodendrocytes (ASPA⁺ EdU⁺) in VTA (saline, n = 12; morphine (10 mg kg⁻¹), $n = 8$; morphine (20 mg kg⁻¹), $n = 11$; cocaine, $n = 16$). **k**, The snRNA-seq identifies baseline interactions among VTA cell populations. DA neurons show strong interactions with OPCs (left, thick line from DA neurons to OPCs) and OPCs show strong interactions with DA neurons (right, thick line from OPCs to DA neurons). **f**, Paired two-tailed $t$-test. **c**,**g**,**l**,**j**, Unpaired two-tailed $t$-test. NS, not significant; $P > 0.5$, *$P < 0.05$, ***$P < 0.001$. Each data point represents a mouse; data shown as mean; error bars, s.e.m. **a**,**d**, Schematics created with BioRender.com.

morphine-evoked oligodendroglial effects observed in vivo are not due to direct effects of the drug but rather mediated through cell–cell interactions (Extended Data Fig. 6a,b). Similarly, we did not detect a change in OPC proliferation in response to varying dopamine concentrations in vitro, indicating that oligodendroglial cells are not simply responding to neurotransmitter release in the VTA. Next, to test for

possible direct effects of morphine mediated by κ-opioid receptors, we conditionally knocked out Oprk1 (κ-opioid receptor) from OPCs to test whether morphine directly influences the oligodendroglial response to morphine in vivo (Extended Data Fig. 6c). We generated *Oprk*$^{lox/lox}$; *Pdgfra-Cre*$^{ERT}$ mice (referred to as *Oprk*$^{OPC-/-}$ hereafter) by crossing inducible *Pdgfra-Cre*$^{ERT}$ mice to floxed *Oprk* (*Oprk*$^{lox/lox}$) mice. Three weeks after deletion of Oprk1 from OPCs, both control and *Oprk*$^{OPC-/-}$ mice exposed to morphine (10 mg kg$^{-1}$) exhibited increased locomotor activity and acquired a strong preference for the morphine-paired conditioning chamber (Extended Data Fig. 6d–f). Both control and *Oprk*$^{OPC-/-}$ mice showed similar numbers of proliferating VTA OPCs (Extended Data Fig. 6g,h), indicating that the OPC proliferative response to morphine is not mediated through Oprk1. Additionally, VTA snRNA-seq indicated that oligodendroglial cells in VTA do not express appreciable amounts of opioid receptor genes (*Oprk1*, *Oprm1* and *Oprd1*) nor dopamine receptor genes (*Drd1–5*), which did not change with morphine-induced reward learning (Extended Data Fig. 6i,j).

## Oligodendrogenesis in drug reward

Neuronal activity-regulated myelination is required for certain forms of learning in the healthy brain[4,6,7]. Because morphine-induced CPP behaviour is a form of associative reward learning, we proposed that oligodendrogenesis in the VTA is necessary for forming associations between morphine and the conditioning chamber context. To test this hypothesis, we genetically blocked oligodendrogenesis by means of inducible deletion of myelin regulatory factor (*Myrf*), a transcription factor essential for oligodendroglial differentiation, in OPCs by crossing inducible *Pdgfra-Cre*$^{ERT}$ mice to floxed *Myrf* (*Myrf*$^{lox/lox}$) mice generating *Myrf*$^{lox/lox}$; *Pdgfra-Cre*$^{ERT}$ (referred to as *Myrf*$^{OPC-/-}$ hereafter). Deletion of *Myrf* in OPCs is achieved upon tamoxifen administration, after which OPCs cannot differentiate into new oligodendrocytes and instead undergo apoptosis, whereas pre-existing oligodendrocytes and myelin remain unaffected[4]. After 15 weeks of *Myrf* loss, we analysed oligodendroglial lineage cell generation in the myelin-rich corpus callosum to determine the baseline effects of *Myrf* deletion (Extended Data Fig. 7a). Within 4 weeks of EdU labelling, a significant reduction in recently generated oligodendroglial lineage cell (EdU$^+$Olig2$^+$) density was apparent (Extended Data Fig. 7b). Therefore, tamoxifen administration at 7 weeks of age, immediately before the CPP assay, prevented further generation of oligodendrocytes in *Myrf*$^{OPC-/-}$ animals during morphine conditioning but this differentiation process central to oligodendrogenesis remained intact in the control Cre$^-$ littermates (Fig. 4a).

In the CPP experiments, *Myrf*$^{OPC-/-}$ animals showed increased locomotor sensitization similar to the controls, yet morphine CPP was significantly abrogated (Fig. 4b–d). As expected, *Myrf*$^{OPC-/-}$ animals exhibit a substantially reduced oligodendroglial response (EdU$^+$Pdgfrα$^+$) in the VTA compared with littermate controls (Fig. 4e,f). Taken together, these findings suggest that oligodendrogenesis is necessary for morphine reward learning but not for locomotor sensitization.

Because *Myrf* loss and consequent abrogation of oligodendrogenesis is linked to impairments in hippocampal memory consolidation over time (at 4 weeks but not at 1 day, after *Myrf* recombination)[6,7], we tested *Myrf*$^{OPC-/-}$ animals to determine whether the observed abrogation of CPP may be a result of general memory deficits by performing a memory task within the timeline that we perform morphine CPP experiments. One week following tamoxifen-induced Myrf deletion, coinciding with the timeframe of the CPP paradigm, we tested mice in the novel object recognition task (NORT), in which mice are introduced to two identical objects and then exposed to one familiar and one novel object 24 h later (Extended Data Fig. 7c). Mice with intact memory and attention recognize the old object and spend more time exploring the novel object. At 24 h, both control and *Myrf*$^{OPC-/-}$ animals explored the novel object for more time than they did the familiar object (Extended Data Fig. 7d). These results demonstrate that, at this timepoint—1 week

after *Myrf* recombination—loss of oligodendrogenesis does not cause significant general memory deficits.

Brain-derived neurotropic factor-tropomyosin receptor kinase B (Bdnf-TrkB) signalling is a key mechanism mediating activity-regulated oligodendrogenesis in frontal cortex and corpus callosum[5]. Proposing that this neuron–OPC signalling pathway is operant in VTA, we conditionally deleted *TrkB* specifically from OPCs by crossing inducible *Pdgfra-Cre*$^{ERT}$ mice to floxed *TrkB* (*TrkB*$^{lox/lox}$) mice generating *TrkB*$^{lox/lox}$; *Pdgfra-Cre*$^{ERT}$ (referred to as *TrkB*$^{OPC-/-}$ hereafter) (Fig. 4g). We administered tamoxifen immediately before morphine conditioning to delete *TrkB* in OPCs and prevent activity-regulated oligodendrogenesis. *TrkB*$^{OPC-/-}$ animals exhibited locomotor sensitization similar to the littermate controls but morphine CPP was significantly abrogated (Fig. 4h–j). The oligodendroglial response (EdU$^+$Pdgfrα$^+$) in the VTA was also substantially reduced compared to the controls (Fig. 4k,l). These results support the *Myrf*$^{OPC-/-}$ mouse model findings described above, demonstrating that oligodendrogenesis is necessary for morphine reward and indicate a potential mechanism (Bdnf–TrkB signalling) by which oligodendroglia can respond to DA neuronal activity. Consistent with these findings, we also found that VTA OPCs expressed high levels of *TrkB* (Extended Data Fig. 5g) and that *Bdnf* is expressed by a variety of neuronal types in the VTA including DA neurons (Extended Data Fig. 5h).

Because prolonged *TrkB* loss in OPCs can cause impaired attention and memory (at 4 weeks after *TrkB* recombination)[5], we evaluated *TrkB*$^{OPC-/-}$ animals using the NORT test to determine whether the observed abrogation of CPP may be a result of general memory deficits within the timeline that we perform morphine CPP experiments (Extended Data Fig. 7c,e). Both control and *TrkB*$^{OPC-/-}$ animals explored the novel object for more time than they did the familiar object (Extended Data Fig. 7e) indicating intact attention and memory function. These results demonstrate that, 1 week after *TrkB* recombination, preventing oligodendrogenesis by blocking Bdnf-TrkB signalling does not cause significant general memory deficits.

We also tested the prolonged effects of blocking oligodendrogenesis by assessing morphine (10 mg kg$^{-1}$) CPP 4 weeks after tamoxifen-induced *Myrf* deletion (Extended Data Fig. 7f). Replicating the data at the 1 week timepoint, both control and *Myrf*$^{OPC-/-}$ animals showed locomotor sensitization to morphine (Extended Data Fig. 7g). Control animals acquired a robust preference for the morphine-paired conditioning chamber whereas for *Myrf*$^{OPC-/-}$ animals this response was significantly abrogated (Extended Data Fig. 7h,i). Consistently, *Myrf*$^{OPC-/-}$ animals showed a substantially reduced oligodendroglial response (EdU$^+$Pdgfrα$^+$) in the VTA (Extended Data Fig. 7j).

To address whether blocking oligodendrogenesis can alter behaviours in response to natural rewards, we tested *Myrf*$^{OPC-/-}$ animals 4 weeks after recombination for sociability and sucrose preference (Extended Data Fig. 7k). In the three-chamber social preference test, we found that both control and *Myrf*$^{OPC-/-}$ animals exhibited comparable preferences towards a same-sex juvenile conspecific over an inanimate object (Extended Data Fig. 7l), reflected in similar amounts of time investigating the juvenile mouse (Extended Data Fig. 7m). These data demonstrate that blocking oligodendrogenesis at this timepoint and age does not affect general sociability. We also tested preference for a natural reward, sucrose, to determine whether the abrogation of a preference for morphine in *Myrf*$^{OPC-/-}$ animals could be because of anhedonia-like effects. Both control and *Myrf*$^{OPC-/-}$ animals showed a strong preference towards sucrose solution over water, indicating that blocking oligodendrogenesis did not cause a general decrease in reward processing (Extended Data Fig. 7n). Taken together these data demonstrate that oligodendrogenesis is required for morphine-induced reward learning but not for intrinsic reward behaviours.

DA neurons also mediate associative learning for natural rewards such as food, and a distinct subpopulation of DA neurons mediate aversion learning[42]. To address whether natural reward or aversion learning promotes oligodendrogenesis, we performed food-induced

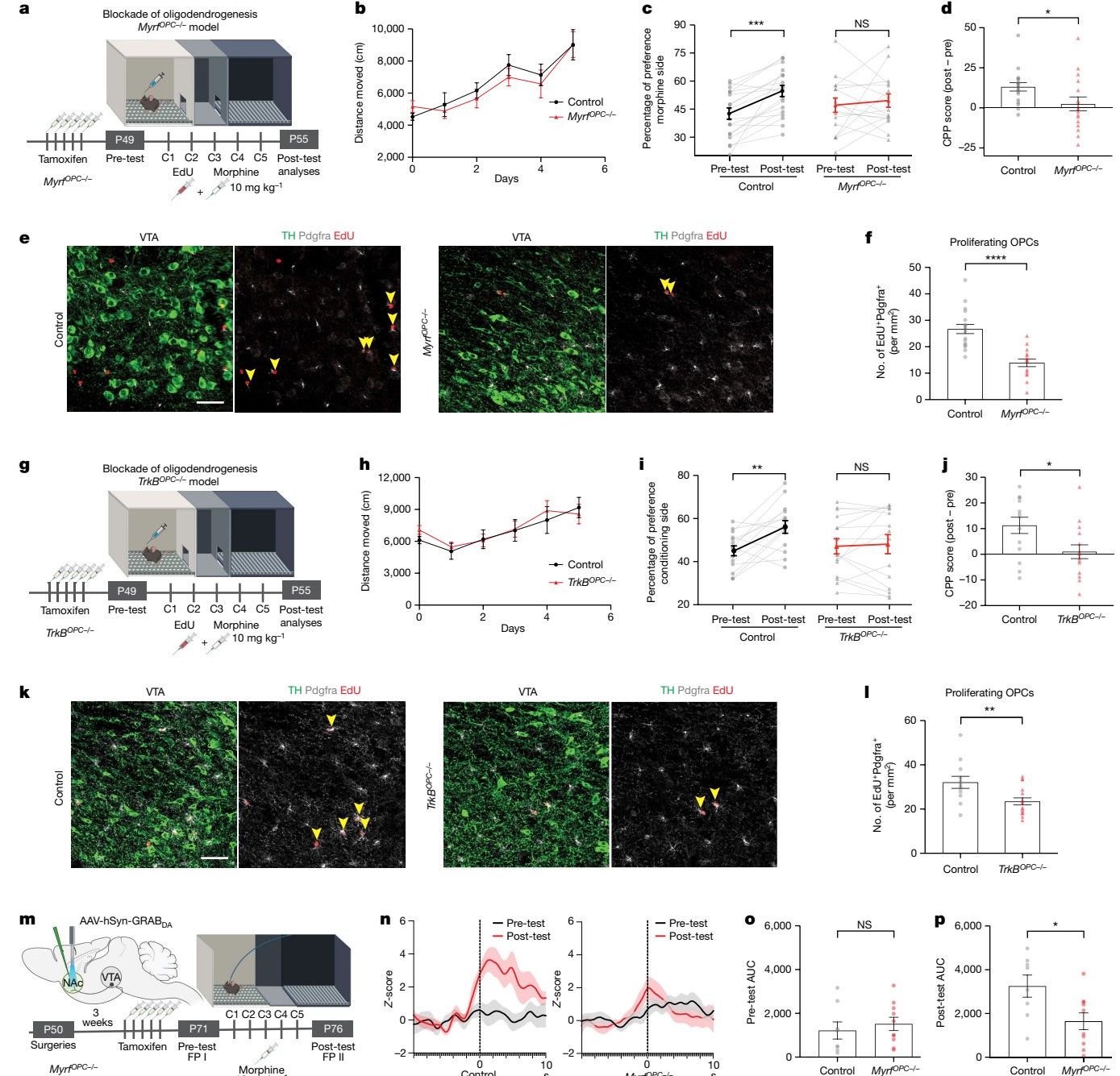

**Fig. 4 | Genetic blockade of oligodendrogenesis abrogates morphine-induced reward learning by attenuating DA release. a**, Experimental paradigm for CPP testing in *Myrf*-wild-type control or *Myrf*$^{OPC-/-}$ mice. **b**, Locomotor sensitivity during morphine conditioning (control, $n = 18$ mice; *Myrf*$^{-/-}$, $n = 16$ mice) (control, $n = 14$; *TrkB*$^{-/-}$, $n = 15$). **c**, Control mice acquire a place preference for the morphine conditioning chamber, whereas *Myrf*$^{OPC-/-}$ mice or *TrkB*$^{OPC-/-}$ mice (**i**) do not show a strong preference (control, $n = 18$; *Myrf*$^{-/-}$, $n = 16$). **d,j**, *Myrf*$^{-/-}$ mice (**d**) and *TrkB*$^{OPC-/-}$ mice (**j**) show decreased CPP score (post-test − pre-test preference). **e**, Representative confocal micrograph of proliferating OPCs in VTA after morphine CPP, arrowheads denote proliferative OPCs (Pdgfrα$^+$EdU$^+$), TH (green), Pdgfrα$^+$ (grey), EdU$^+$ (red). Scale bar, 50 µm. **f**, Proliferative OPCs in VTA of control or *Myrf*$^{OPC-/-}$ mice after morphine CPP ($n = 18$ control mice; $n = 15$ *Myrf*$^{OPC-/-}$ mice) **g**, Experimental paradigm for CPP in *TrkB*$^{OPC-/-}$ mice.

**h**, Locomotor sensitivity during morphine conditioning in control or *TrkB*$^{OPC-/-}$ mice. **i,j,k**, CPP as in **c** (**i**), **d** (**j**) and **e** (**k**) in the *TrkB*$^{OPC-/-}$ mouse model ($n = 14$ control mice; $n = 15$ *TrkB*$^{OPC-/-}$). **l**, as in **f**, $n = 12$ control; $n = 16$ *TrkB*$^{-/-}$ mice. **m**, Experimental paradigm for dopamine release detection in *Myrf*$^{OPC-/-}$ model with fibre photometry (FP) during before and after morphine CPP. **n**, Group average GRAB$_{DA}$ responses in NAc on first entry into the conditioning chamber in control and *Myrf*$^{OPC-/-}$ mice. **o,p**, Dopamine release in control and *Myrf*$^{OPC-/-}$ mice before (**o**) and after (**p**) morphine CPP ($n = 8$ control mice; $n = 10$ *Myrf*$^{OPC-/-}$). **c,i**, Paired two-tailed *t*-test. **d,f,j,l,o,p**, Unpaired two-tailed *t*-test. NS, not significant; $P > 0.5$, *$P < 0.05$, **$P < 0.01$, ***$P < 0.001$, ****$P < 0.0001$. Each data point represents a mouse; data shown as mean; error bars, s.e.m. **a,g,m**, Schematics created with BioRender.com.

CPP or naloxone-induced conditioned place aversion testing, as above. We found that neither food reward nor naloxone aversion induced oligodendrogenesis and neither were affected by genetic blockade of oligodendrogenesis (Extended Data Fig. 8), indicating that although oligodendrogenesis is necessary for drug-evoked learning, it does not affect food reward or naloxone-induced aversion learning.

## Oligodendrogenesis tunes DA release in NAc

Adaptive myelin changes promote neural network coordination to ensure proper spike-time arrival for optimal network function[7,22,23]. We proposed that morphine-induced oligodendrogenesis tunes DA circuitry, thereby optimizing dopamine release dynamics in projection targets to facilitate reward learning and memory retrieval. To test this possibility, we expressed the genetically encoded fluorescent dopamine sensor GRAB$_{DA}$ (ref. 43) in the NAc medial shell and recorded real-time endogenous dopamine dynamics in mice undergoing the morphine CPP procedure. We performed recordings during the pre-test and post-test to determine whether dopamine release is modified with morphine reward learning (Fig. 4m and Extended Data Fig. 9a,b). In line with our previous findings (Fig. 4), both control and $Myrf^{OPC-/-}$ animals showed locomotor sensitization over the morphine conditioning days and control animals acquired a strong preference to the morphine-paired conditioning chamber which was absent in $Myrf^{OPC-/-}$ animals (Extended Data Fig. 9c,d). During the pre-test baseline recording before reward conditioning, both control and $Myrf^{OPC-/-}$ animals showed small spontaneous NAc dopamine transients during exploration of the chamber to be paired with morphine, probably due to novelty (Fig. 4n,o). After 5 days of conditioning with morphine, control animals showed a significant increase in NAc dopamine concentrations on their first entry into the morphine-paired chamber during the post-test, an expression of reward expectation for the previous morphine experience associated with the chamber[44] (Fig. 4n,p). By contrast, $Myrf^{OPC-/-}$ animals showed a significantly blunted dopamine response on entry, similar to their pre-test concentrations, suggesting an absence of an association between the chamber and the previous morphine rewards (Fig. 4n,p). During the pre-test, dopamine release in $Myrf^{OPC-/-}$ animals was similar to controls, illustrating that $Myrf^{OPC-/-}$ animals do not exhibit lower baseline concentrations of dopamine (Fig. 4o). After morphine conditioning, control animals associate the conditioning chamber with the morphine reward and exhibit a significant increase in NAc dopamine concentrations when they enter into the conditioning chamber with a reward expectation during the post-test (Fig. 4n). This learned dopamine response is lacking in oligodendrogenesis-deficient $Myrf^{OPC-/-}$ animals, reflected in the difference score of fluorescence from the post-test and pre-test (Fig. 4n,p and Extended Data Fig. 9e). Overall, these results indicate that morphine-induced oligodendrogenesis in the VTA promotes dopamine release dynamics in the NAc, critical for morphine reward learning.

## Discussion

Mounting evidence indicates that experience-dependent changes in myelin contribute to proper neural circuit function and brain health. Here we show that DA neuronal activity induces myelin plasticity on DA axons in the reward system in a circuit- and region-specific manner, affecting proximal axonal segments of DA neurons within VTA. Such myelin changes closer to the axonal initial segment[45] can exert profound changes in conduction velocity[46,47] and consequently on circuit function. The same oligodendroglial responses are elicited by optogenetic stimulation or disinhibition of DA neurons or by drugs of abuse, such as morphine and cocaine. These effects are not explained by direct effects of dopamine or morphine on OPCs. By contrast, TrkB signalling in OPCs is required for morphine-induced oligodendrogenesis in the VTA, implicating Bdnf-TrkB signalling in DA neuron-to-OPC interactions promoting these myelin changes. Our findings are most consistent with de novo myelination; whether myelin remodelling also occurs in reward circuitry and whether the observed oligodendroglial plasticity contributes to myelin-independent oligodendroglial functions, remain open questions.

Myelin plasticity in the healthy brain has been predicted[22,23] and experimentally shown[7] to tune neural circuits for optimal function. Myelin plasticity-mediated circuit adaptations may synchronize neuronal activity across the circuit and increase spatiotemporal precision—such as coordinated or enhanced dopamine release in NAc—to mediate reward-related behaviours. Concordantly, we find that the myelin plasticity observed in VTA is required for morphine-induced reward learning but not for food reward learning or aversion learning and regulates dopamine release dynamics in NAc. This underscores a critical role for activity-regulated myelin changes in the function of reward circuitry relevant to drugs of abuse. Why neither food reward nor aversion learning elicit an oligodendroglial response remains an open question; possible explanations requiring future testing include differential strength of various stimuli or differential effects of distinct subpopulations of DA neurons.

Reward learning is important for healthy interactions of animals with their environment and survival but can also contribute to pathological substance-use disorders. Although a single dose of a drug is not sufficient to cause a substance-use disorder, the resulting circuit alterations may lay the foundation for stronger and more persistent modifications which enable development of substance-use disorder, such as strong drug-associated memories and high sensitivity to drug cues. Understanding these persistent changes is crucial for elucidating mechanisms that give rise to and maintain addictive behaviours. Here we show that morphine-induced DA neuronal activity drives myelination of reward circuitry in the VTA, which in turn regulates morphine-seeking behaviours. This raises the possibility that adaptive myelination may become maladaptive in the context of opioid reward after repeated cycles of intoxication and withdrawal, thereby promoting the development of opioid use disorder. The findings here bring into focus another dimension of drug-evoked plasticity by defining new players in the neural circuit modifications that lead to addiction. Maladaptive myelination in the VTA could represent a key neural substrate of this pathological learning, suggesting myelination as a potential therapeutic target for opioid use disorder.

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

## Methods

### Animals

All procedures were performed in accordance with guidelines set in place by National Institutes of Health and approved by the Stanford University Institutional Care and Use Committee.

Both male and female mice were used equally in all experiments and were randomized to treatment conditions. Hemizygous *Dat-Cre* (the Jackson Laboratory, 006660) mice were bred with C57BL/6 J (the Jackson Laboratory, 000664). Optogenetic experiments were performed on animals hemizygous for *Dat-Cre* or homozygous for *Vgat-Cre* (the Jackson Laboratory, 028862). For oligodendrocyte specific expression of mGFP, homozygous mT/mG$^{lox/lox}$ (the Jackson Laboratory, 007676) animals were crossed with *Plp1-CreER*$^T$ (the Jackson Laboratory, 005975). For conditional deletion of *Myrf*, hemizygous *Pdgfra-CreER*$^{TM}$ (the Jackson Laboratory, 018280) mice were bred with homozygous *Myrf*$^{lox/lox}$ mice[48] (the Jackson Laboratory, 010607) to generate hemizygous *Pdgfra-CreER*$^{TM}$ and heterozygous *Myrf*$^{lox/+}$, which were backcrossed to homozygous *Myrf*$^{lox/lox}$ to achieve hemizygous *Pdgfra-CreER*$^{TM}$ and homozygous *Myrf*$^{OPC−/−}$ (*Myrf*$^{lox/lox}$; *Pdgfra-CreER*$^{TM}$). Littermates that lacked *Pdgfra-CreER*$^{TM}$ were used as control animals. For conditional deletion of *TrkB*, hemizygous *Pdgfra-CreER*$^{TM}$ (the Jackson Laboratory, 018280) mice were bred with homozygous *TrkB*$^{lox/lox}$ (MMRC, 033048-UCD) to generate hemizygous *Pdgfra-CreER*$^{TM}$ and heterozygous *TrkB*$^{lox/+}$, which were backcrossed to homozygous *TrkB*$^{lox/lox}$ to achieve hemizygous *Pdgfra*-CreER$^{TM}$ and homozygous *TrkB*$^{OPC−/−}$ (*TrkB*$^{lox/lox}$; *Pdgfra-CreER*$^{TM}$), which was referred to as OPC-TrkB cKO in a previous publication[5]. Littermates that lack *Pdgfra-CreER*$^{TM}$ were used as control animals. For conditional deletion of *Oprk*, hemizygous *Pdgfra-CreER*$^{TM}$ (the Jackson Laboratory, 018280) mice were bred with homozygous *Oprk1*$^{lox/lox}$ (the Jackson Laboratory, 030076) to generate hemizygous *Pdgfra-CreER*$^{TM}$ and heterozygous *Oprk1*$^{lox/+}$, which were backcrossed to homozygous *Oprk*$^{lox/lox}$ to achieve hemizygous *Pdgfra-CreER*$^{TM}$ and homozygous *Oprk1*$^{lox/lox}$ (*Oprk1*$^{lox/lox}$; *Pdgfra-CreER*$^{TM}$). Littermates that lack *Pdgfra-CreER*$^{TM}$ were used as control animals. To initiate Cre-dependent deletion of *Myrf*, *TrkB* or *Oprk*, animals were intraperitoneally injected with 100 mg kg$^{-1}$ tamoxifen (Sigma-Aldrich) for 5 days at 7 weeks or 4 weeks of age. For mGFP expression mT/mG animals were injected with the same concentration of tamoxifen for 3 days at 4 weeks of age. For testing effects of drugs on oligodendroglial cells, wild-type C57BL/6 J (the Jackson Laboratory, 000664) mice were used. All animals were housed in a 12 h light/dark cycle with unrestricted access to food and water. Behavioural experiments were performed during the same circadian period (07:00–19:00). All mouse experiments were repeated in at least two independent experiments conducted at different times. Sample sizes were guided by power calculations.

### Viral vectors

For optogenetics experiments, Dat-Cre-dependent expression of channelrhodopsin was achieved by injection of AAV-DJ-EF1a-DIO-hChR2(H134R)::eYFP (virus titre $1.3 \times 10^{12}$ genome copies per millilitre), Vgat-Cre-dependent expression of halorhodopsin was achieved by injection of AAV-DJ-EF1a-DIO-NpHR3.0::eYFP (virus titre $1 \times 10^{12}$ genome copies per millilitre) and eYFP control by injection of AAV-DJ-EF1a-DIO-eYFP (virus titre $1.6 \times 10^{12}$ genome copies per millilitre). All viral vectors for optogenetics experiments were obtained from Stanford University Gene Vector and Virus Core. For fibre photometry experiments, Grab$_{DA}$ expression was achieved by injection of AAV9-hSyn-DA2m (DA4.4) (virus titre, $1.64 \times 10^{13}$ genome copies per millilitre) obtained from WZ Biosciences[49].

### Surgical procedures

Animals were anaesthetized with 1–4% isoflurane and placed in a stereotaxic apparatus. For optogenetic stimulation experiments, 1 µl of AAV-ChR2::eYFP or AAV-eYFP viral vectors were unilaterally injected using Hamilton Neurosyringe and Stoelting stereotaxic injector over 5 min. Coordinates for viral injections and optic ferrule placements were measured from bregma. Viral vectors were injected into VTA at coordinates, anterior–posterior (AP) = −2.8 mm, mediolateral (ML) = −0.3 mm, dorsoventral (DV) = −4.4 mm and an optic ferrule was placed at AP = −2.8 mm, ML = −0.3 mm, DV = −4.00 mm. For NAc stimulations an optic ferrule was placed at AP = +1.25 mm, ML = −0.75 mm, DV = −4.00 mm.

### Optogenetic stimulations

Optogenetic stimulations were performed at least 3 weeks after the viral vector delivery and 1 week after optic ferrule implantation. Freely moving animals were connected to a 473 nm diode-pumped solid-state laser system with a monofibre patch cord. Phasic DA neuron stimulation, for both neuronal cell bodies and axon terminals, was performed at 30 Hz, with eight 5 ms pulses of 473 nm light delivery every 5 s at a light power output of 15 mW (about 475 mW mm$^{-2}$) from the tip of the optic fibre (200 µm core diameter, numerical aperture = 0.22, Doric Lenses)[31]. Tonic DA neuron stimulation was performed at 1 Hz, with 24 15 ms pulses of 473 nm light delivery every 1 min (ref. 30). Acute optogenetic stimulation session lasted for 30 min for phasic or tonic stimulations. Animals were injected intraperitoneally with 40 mg kg$^{-1}$ of EdU (Invitrogen, E10187) before and after the session and perfused 3 h after the start of the stimulation. Chronic phasic optogenetic stimulations were performed for 10 min a day, for 7 days, with EdU injections for all stimulation sessions. Animals were perfused 4 weeks after the cessation of stimulations. For inhibition experiments, freely moving animals were connected to a 595 nm high-power LED system with a monofibre patch cord. NpHR inhibition of GABAergic neuron cell bodies was performed with constant 595 nm light delivery at 1 Hz, for 8 s every 10 s at light power output of 15 mW (about 475 mW mm$^{-2}$) from the tip of the optic fibre.

### Fibre photometry

AAV9-hSyn-DA2m (DA4.4) was injected into NAc (AP = +1.25 mm, ML = −0.75 mm, DV = −4.40 mm) and an optic fibre (400 µm core diameter, NA = 0.48, Doric Lenses) was placed just above the injection site (AP = +1.25 mm, ML = −0.75 mm, DV = −4.30 mm). Mice were administered with tamoxifen 1 day to 4 weeks before starting CPP. After allowing 3 to 4 weeks for viral expression, Grab$_{DA}$ signal was recorded during the pre-test while the mice explored each chamber of the three-chamber CPP apparatus for 30 min. Mice then went through morphine conditioning for 5 days. The Grab$_{DA}$ signal was then recorded again during post-test exploration for 30 min, in an identical manner to the pre-test.

Fibre photometry data were acquired with Synapse software controlling an RZ5P lock-in amplifier (Tucker-Davis Technologies). Grab$_{DA}$ was excited by frequency-modulated 465 and 405 nm LEDs (Doric Lenses). Optical signals were band-pass filtered with a fluorescence mini cube FMC4 (Doric Lenses) and signals were digitized at 6 kHz. Signal processing was performed with custom scripts in MATLAB (MathWorks). Briefly, signals were debleached by fitting with a mono-exponential decay function and the resulting fluorescence traces were *Z*-scored. Videos were analysed for the first entry into the morphine conditioning chamber for each animal for the pre-test and post-test. Peristimulus time histograms were constructed by taking the average of 20 s epochs of fluorescence consisting of 10 s before and 10 s after the chamber entry, which is defined as time = 0. Before averaging, each epoch was offset such that the *Z*-score averaged from −10 to −1 s equalled 0. Area under the curve (AUC) was defined as the integral between 0 and +10 s. The AUC difference score was calculated as the (post-test AUC − pre-test AUC) for each animal.

## Behavioural analysis

**Real-time place preference.** Real-time place preference was performed in a two-chamber acrylic box (each chamber $45 \times 20 \times 25\ cm^3$) without any more contextual cues. Each mouse was connected to a 473 nm laser system with a monofibre patch cord and gently placed into the middle section of the cage at the beginning of the test. One chamber of the cage was randomly assigned for optogenetic stimulation before the test. Optogenetic stimulation was turned on every time a mouse entered into the pre-assigned chamber and turned off if the mouse moved to the other chamber. Each session lasted for 20 min. Ethovision XT software (Noldus) was used to determine the time animals spent in each chamber after the test in an automated and condition-blinded manner.

**Conditioned place preference.** The experimental protocol was adapted from the previously described method[35]. CPP was performed in an acrylic rectangular three-chamber apparatus with two distinct chambers which are connected by a neutral corridor. The 'grid chamber' included a 3-mm-wide grid in the floor texture and black stripes over white walls. The 'spots chamber' included a floor with 10-mm-diameter holes and white walls. Both chambers were cubes ($18 \times 18 \times 18\ cm^3$) and connected by a 10-cm-wide corridor ($10 \times 18 \times 18\ cm^3$). The entrance to each grid or spots chamber could be closed off with a transparent acrylic divider. On day 1, mice were tested for baseline preference towards either chamber by placing them in the cage. After 30 min of free exploration between chambers with two different kinds of flooring, mice were returned to their home cage. Mice that show strong preference (more than 75%) to either chamber during pre-test were excluded from further analysis. On the morning of day 2 (conditioning day 1), all mice were administered with saline intraperitoneally and placed in the non-conditioning chamber with either kind of flooring for 15 min and returned to their home cage. The morning session was followed by conditioning afternoon session after 4 h. In the afternoon session, the first group of mice received saline and EdU ($40\ mg\ kg^{-1}$) injections intraperitoneally and were placed in the conditioning chamber. A second group of mice were administered with $10\ mg\ kg^{-1}$ of morphine and EdU ($40\ mg\ kg^{-1}$) intraperitoneally and placed in the conditioning chambers for 30 min. A third group received $20\ mg\ kg^{-1}$ of morphine and EdU ($40\ mg\ kg^{-1}$) intraperitoneally and were placed in the conditioning chambers for 30 min. A fourth group of mice received $15\ mg\ kg^{-1}$ of cocaine and EdU ($40\ mg\ kg^{-1}$) intraperitoneally and were placed in the conditioning chambers for 20 min. For naloxone experiments, animals were injected with $5\ mg\ kg^{-1}$ of naloxone (Tocris-0599) during conditioning sessions. These conditioning sessions were repeated on the following days 3, 4, 5 and 6 (total conditioning days 1–5). On day 7, mice were again allowed to freely move around the cage for 30 min during post-test. Food CPP was performed in a similar manner to drug CPP. Food reward group and food restricted control group were food restricted down to 85% body weight 48 h before the start of CPP and maintained at that weight for the duration of the test. Mice were habituated to the chocolate sucrose pellets in the home cage during the two nights preceding CPP. All animal groups were allowed 30 min of free exploration of the apparatus during a pre-test. Both the spots and grid chambers contained empty, identical weighing boats taped to the floor in the back corner. On the morning of day 2 (conditioning day 1), mice were placed into the non-conditioning chamber with an empty weighing boat for 30 min. No saline injections were administered. In the afternoon, mice were administered EdU ($40\ mg\ kg^{-1}$) intraperitoneally and placed in the conditioning chamber with a weighing boat full of chocolate sucrose pellets for food reward group and empty weighing boats for food restricted control group for 30 min. Food reward mice were allowed to freely consume sucrose pellets throughout the conditioning session. These conditioning sessions were repeated over the next 4 days. On day 7, mice were allowed to freely move around the apparatus during a 30 min post-test, in which both the spots and grid chambers contained fresh weighing boats with no sucrose pellets. All drug conditioning sessions were recorded for capturing locomotion data. However, the initial CPP experiments which are used for the histological data were performed during the Covid-19 pandemic lockdown conditions, so we were not able to record the conditioning sessions of most experimental groups. Afterwards we ran more CPP assays to record the locomotion data and CPP preference, which resulted in different numbers of animals across these analysis in Fig. 3. The time spent on each chamber with either type of grid and the distance travelled by each mouse were analysed using Ethovision XT software (Noldus) in an automated and condition-blinded manner.

**Novel-object recognition.** The experimental protocol was adapted from the previously described method[50]. Animals were handled daily for the week leading up to the test for 5 min each day and habituated to the experimental room. Mice were placed in the acrylic experimental cage ($50 \times 50 \times 50\ cm^3$) to acclimatize for 10 min on the day before testing. On the day of the testing, mice were tested for anxiety by placing into the experimental cage for 10 min and video recorded. If an animal spent less than 2 min in the centre of the box (10 cm away from the walls), it was regarded as too anxious for the test and discarded from analysis. Mice were then placed in the home cage for 5 min. For the training phase, mice were placed in the experimental chamber with two identical inanimate objects (about 5 cm in size). Each time the mouse was placed in the experimental chamber it was facing away from the objects towards the opaque walls. The mouse was allowed to explore these objects for 10 min, then was returned to its home cage. Both the experimental chamber and the objects were cleaned using 70% ethanol. For the novel object testing phase, which is performed 24 h after the training phase, the mouse was returned to the cage to explore these objects for 10 min. One of these objects was returned to the experimental cage and a novel inanimate object of similar size was placed into the cage. The objects used as novel and familiar and the position of the novel object were counterbalanced from trial to trial, animal to animal. All the objects used were initially tested to ensure there was no bias or preference for the animals. All sessions were camera recorded and analysed using Ethovision XT software (Noldus) in an automated and condition-blinded manner. Any exploratory head gesturing within 2 cm of the object, including sniffing and biting, was considered as investigation but climbing onto the object was not considered. Only animals that explored the objects for a minimum of 20 s were included in the analysis.

**Sucrose preference.** The experimental protocol was adapted from the previously described method[51]. Mice were tested for sucrose preference 4 weeks after the induction of Myrf deletion by tamoxifen administration. On the first day, animals were habituated individually in experimental cages and to drinking bottles with water. Next day, one water bottle and one sucrose solution bottle (1% wt/vol) were provided for ad libitum access to mice. Water and sucrose solution were replenished every 24 h and consumption was determined per day over 3 days. Positions of the sucrose and water bottles were swapped daily to prevent a bias towards bottle location. All measurements were corrected for spillage by subtracting the volume loss from a control bottle.

**Three-chamber social interaction.** Mice were tested for social preference 4 weeks after the induction of *Myrf* deletion by tamoxifen administration. Animals were habituated to the experimental room and the three-chamber experimental cage before test. The three-chamber acrylic experimental cage (each chamber $45 \times 20 \times 25\ cm^3$) contained two metal grid pencil cups (10 cm in diameter) in each of the outer chambers. The chambers were divided with transparent acrylic walls with 15-cm-wide entrance holes. One of the chambers contained a novel same-sex juvenile in the pencil cup and the other chamber contained an inanimate object in the pencil cup. The test mouse was placed into the

middle chamber and 1 min after the chamber dividers were removed and the mouse was allowed to explore each chamber freely for 20 min. The locations of the novel mouse and the inanimate object were counterbalanced between sessions. The time spent on interacting with either pencil cup was analysed using Ethovision XT software (Noldus) in an automated and condition-blinded manner.

## Brain tissue processing

Mice (saline control, $n = 3$; 10 mg kg$^{-1}$ of morphine, $n = 3$) were subjected to CPP test as described above. On day 7, mice were again allowed to move freely around the cage for 30 min and time spent on each chamber with either type of grid was analysed. After the test, the mice were transcardially perfused with 20 ml of perfusion buffer (110 mM NaCl, 10 mM HEPES, 25 mM glucose, 75 mM sucrose, 7.5 mM MgCl$_2$, 2.5 mM KCl, 5 µg ml$^{-1}$ of actinomycin D and 10 µM triptolide in UltraPure DNase/RNase-free distilled water). The brains were promptly collected, flash frozen in isopentane on dry ice and stored at −80 °C. Frozen brains were sectioned inside a cryostat until AP = −2.8 mm and VTA from both hemispheres were punched out using a 1.75 mm biopsy punch with about 1 mm depth. Punches were stored at −80 °C for processing.

## Isolation and sorting of nuclei

Punches for each mouse were added to a Wheaton Dounce homogenizer containing 1 ml of ice-cold nuclei isolation medium (10 mM Tris pH 8.0, 250 mM sucrose, 25 mM KCl, 5 mM MgCl$_2$, 0.1% Triton X-100, 1% RNasin Plus, 1× protease inhibitor, 0.1 mM DTT, 5 µg ml$^{-1}$ of actinomycin D, 10 µM triptolide and anisomycin in UltraPure DNase/RNase-free distilled water). Tissues were dissociated by ten strokes with the loose Dounce pestle followed by ten strokes with the tight pestle. Homogenates were passed through a 40 µm cell strainer and centrifugated at 900g for 15 min to pellet the nuclei. Nuclei were resuspended in 1 ml of resuspension buffer (1× phosphate buffer saline (PBS), 1% nuclease free BSA and 0.5% RNasin Plus) and centrifuged at 900g for another 15 min to pellet the nuclei. Supernatant was removed and nuclei were then stained by adding 500 µl of resuspension buffer containing 0.1 µg ml$^{-1}$ of DAPI and transferred to FACS tubes. Nuclei were then FACS sorted using a Sony MA900 sorter with a 70 µm chip. A standard gating strategy was applied to all samples. First, nuclei were gated on their size and scatter properties. Doublet discrimination gates were used to exclude aggregates of nuclei. Lastly, nuclei were gated on DAPI. Single nuclei were sorted into chilled PCR tubes containing 10 µl of resuspension buffer. The counts of nuclei were verified using a haemocytometer.

## RNA-seq library preparation and sequencing

10X Chromium Next GEM Single Cell 3′ HT Kit v.3.1 was used for library preparation. Following manufacturer's instructions, nuclei were promptly mixed with the master mix and loaded onto a 10X Chromium Next GEM Chip M and ran on a 10X Chromium X instrument aiming to recover 20,000 cells from each. We generated 10X 3′ RNA-seq libraries following the manufacturer's protocols. After complementary DNA amplification and after library construction, quality controls were performed to ensure quality of samples. Libraries were loaded at 650 pM along with 1% PhiX control on an Illumina NextSeq2000 and paired-end sequenced (28 cycles read1, 10 cycles i7 index, 10 cycles i5 index, 90 cycles read2) at a targeted depth of 20,000 reads per nucleus.

## RNA-seq data analysis

Raw read (FASTQ) files were aligned to the mouse reference genome (mm10-2020-A) using Cell Ranger (v.7.0.1). For snRNA-seq, all analysis, quantification and statistical testing was completed using R (v.4.2.2). Seurat (v.4.3.0) was used for preprocessing, dimensionality reduction, clustering and differential expression testing. Cell Ranger count matrices for each sample were merged and low-quality cells with fewer than 200 detected genes or greater than 5% mitochondrial mapping unique molecular identifiers (UMIs) and putative doublets with greater than 4,000 detected genes were removed. A total of 80,302 nuclei passed these quality control criteria (39,780 saline and 40,522 morphine). UMI count data were then normalized, such that the counts per each gene per cell were divided by the total UMIs per cell, multiplied by a scale factor of 10,000 and natural log transformed. Variance stabilizing transformation was performed and the top 2,000 variable genes were identified, followed by principal component analysis. UMAP was conducted using the first 12 principal components and graph-based clustering was used to identify clusters with a resolution parameter of 0.8. To focus on oligodendroglial lineage cells, oligodendrocytes and OPCs were then subsetted and reclustered using similar parameters. Trajectory from OPCs to mature oligodendrocytes were labelled using previously defined markers for oligodendroglial subpopulations[37]. All differential expression testing was performed using Wilcoxon rank-sum tests with Bonferroni corrections for several comparisons. Adjusted $P < 0.05$ with accompanying average log$_2$-fold changes greater than 0.25 in magnitude were considered statistically significant. ScType and differential expression testing and were first performed to identify cell types based on canonical marker gene expression and clusters of the same cell type were merged[52]. Gene set enrichment analysis was performed using Cluster Profiler (v.4.6.2) to identify enrichment of gene ontology biological process terms in cluster differentially expressed genes using a hypergeometric test. Cell–cell communication analysis and inference was performed using CellChat (v.1.6.1) to calculate the aggregated cell–cell communication networks and identify signals contributing to outgoing or incoming signalling of different cell groups[38].

## In vitro OPC analysis

To test the direct effects of morphine and dopamine, in vitro OPC proliferation assay was used. C57BL/6 P4-5 mice pups were rapidly decapitated and brains were processed in Hibernate-A medium (Thermo Fisher Scientific, A12475-01). Resulting tissue was enzymatically disassociated in buffer containing HEPES-HBSS with DNase (Worthington Biochemical LS002007) and Liberase (Roche Applied Sciences 05401054001) at 37 °C on a rotator. Tissue mixture then was triturated with 1,000 µl tip and passed through a 100 µm cell strainer. OPCs were isolated using the CD140 (Pdgfrα) Microbead kit (MACS, Miltenyi Biotex 130-101-502) according to the manufacturer's instructions. A total of 30,000 cells were seeded per well, on laminin-coated (Thermo Fisher Scientific, 23017015) coverslips in a 24-well plate. OPC media containing DMEM (Thermo Fisher Scientific, 11320082), glutamax (Invitrogen, 35050-061), sodium pyruvate (Invitrogen, 11360070), MEM non-essential amino acids (Thermo Fisher Scientific, 11140076), antibiotic-antimytotic (GIBCO), N21-MAX (R&D systems, AR012), trace elements B (Corning, 25-022-Cl), 5 mg ml$^{-1}$ of *N*-acetyl cysteine (Sigma-Aldrich, A9165), 10 ng ml$^{-1}$ of PDGFAA (Shenandoah Biosciences, 200-54), 10 ng ml$^{-1}$ of CNTF (PeproTech, 450-13) and 1 ng ml$^{-1}$ of NT-3 (PeproTech, 450-03) was used. For proliferation studies, after 3 days in proliferative media, cells were treated with various concentrations of morphine or dopamine in OPC media for 24 h. In the last 4 h of this treatment, 10 µM EdU was added to the media to label dividing cells. Afterwards, cells were fixed in 4% paraformaldehyde (PFA) for 20 min and incubated in HBSS until immunohistochemistry. All in vitro experiments were performed in triple wells (technical replicate) and independently replicated (biological replicates).

## Immunohistochemistry

All mice were anaesthetized with intraperitoneal injections of 2.5% Avertin (tribromoethanol; Sigma-Aldrich, T48402) and transcardially perfused with 20 ml of 0.1 M PBS. Brains were postfixed in 4% PFA overnight at 4 °C before cryoprotection in 30% sucrose solution for 48 h. For sectioning, brains were embedded in optimum cutting temperature (Tissue-Tek) and sectioned coronally at 40 µm using a sliding microtome (Leica, HM450). For immunohistochemistry, brain sections were stained using the Click-iT EdU cell proliferation

kit (Invitrogen, C10339) according to manufacturer's protocol. Tissue sections were then stained with antibodies following an incubation in blocking solution (3% normal donkey serum, 0.3% Triton X-100 in tris-buffered saline (TBS)) at room temperature for 30 min. Goat anti-Pdgfrα (1:500; R&D Systems, AF1062), rabbit anti-Olig2 (1:500; Abcam, 7349), rabbit anti-ASPA (1:250; EMD Millipore, ABN1698), rat anti-MBP (1:200; Abcam ab7349), rabbit anti-tyrosine hydroxylase (1:500; Millipore Sigma, AB152), mouse anti-tyrosine hydroxylase (1:250; Novus Biologicals, MAB7566), chicken anti-GFP (1:1,000; Aves Labs, GFP-1020), chicken anti-mCherry (1:1,000; ab205402) or rabbit anti-cfos (1:500; Santa Cruz Biotechnology, sc-52) were diluted in 1% blocking solution (1% normal donkey serum in 0.3% Triton X-100 in TBS) and incubated overnight at 4 °C. All antibodies have been validated in the literature for use in mouse immunohistochemistry. To further validate the antibodies, we confirmed that each antibody stained in the expected cellular patterns and brain-wide distributions (for example, nuclear Olig2 staining, cell membrane Pdgfrα staining, tyrosine hydroxylase staining in midbrain DA neurons). The following day, brain sections were rinsed three times in 1× TBS and incubated in secondary antibody solution for 2 h at the room temperature. All secondary antibodies were used at 1:500 concentration including Alexa 488 anti-rabbit (Jackson ImmunoResearch; 711-545-152), Alexa 488 anti-mouse (Jackson ImmunoResearch; 715-545-150), Alexa 488 anti-chicken (Jackson ImmunoResearch; 703-545-155), Alexa 594 anti-chicken (Jackson ImmunoResearch; 703-585-155), Alexa 647 donkey anti-goat (Jackson ImmunoResearch; 705-605-147), Alexa 647 anti-rat (Jackson ImmunoResearch; 712-605-150), Alexa 647 anti-rabbit (Jackson ImmunoResearch; 711-605-152), Alexa 647 donkey anti-goat (Jackson ImmunoResearch; 705-605-147). Sections were then rinsed three times in 1× TBS and mounted with ProLong Gold (Life Technologies, P36930).

### Fluorescence microscopy and quantification

All image analyses were performed by experimenters blinded to the experimental conditions or genotype. EdU stereology images were taken using a Zeiss AxioObserver upright fluorescence microscope with automated stage and tile-scanning capability (Stereo Investigator, Microbrightfield) with ×20 objective. Brain tissue that was damaged during perfusion or tissue processing was excluded from histological analysis. Imaging and cell counting were performed on every sixth 40 μm slice, throughout the extend of DA neurons in the midbrain and extend of NAc. Cells surrounding ferrule-induced tissue damage were not included. Regions with tyrosine hydroxylase labelling (or Dat-Cre»GFP positive for optogenetic studies) were marked and EdU cells in these regions were counted manually using Stereo Investigator software. Resulting numbers were reported as a function of the area (cells per mm$^2$). Higher resolution imaging was conducted by acquiring z-stacks using a Zeiss LSM710 or LSM980 confocal microscope (Carl Zeiss). For MBP intensity analyses, z-stack images were taken of VTA using confocal microscopy with the same laser and fluorescence settings across animals and sections. Mean MBP intensity of pixels over VTA, which is labelled by Dat-Cre»GFP, was quantified using Fiji and reported as a function of the area.

### Electron microscopy and quantification

Four weeks after cessation of chronic optogenetic stimulations, mice were euthanized by transcardial perfusion with Karnovsky's fixative: 2% glutaraldehyde (EMS 16000) and 4% PFA (EMS 15700) in 0.1 M sodium cacodylate (EMS 12300), pH 7.4. For regional accuracy and consistency, brains were sectioned into 250-μm-thick slices on a vibratome (Leica VT1000S) until midbrain. Then 2-mm-diameter tissue punches were collected from VTA region under the optic fibre mark. The samples were then postfixed in 1% osmium tetroxide (EMS 19100) for 1 h at room temperature, washed three times with ultrafiltered water, then en bloc stained for 2 h at room temperature. Samples were dehydrated

in graded ethanol (50%, 75% and 95%) for 15 min each at 4 °C; the samples were then allowed to equilibrate to room temperature and were rinsed in 100% ethanol twice, followed by acetonitrile for 15 min. Samples were infiltrated with EMbed-812 resin (EMS 14120) mixed 1:1 with acetonitrile for 2 h followed by 2:1 EMbed-812:acetonitrile for 2 h. The samples were placed into EMbed-812 for 2 h, then placed into TAAB capsules filled with fresh resin, which were then placed into a 65 °C oven overnight. Sections were taken between 40 and 60 nm on a Leica Ultracut S (Leica) and mounted on 100-mesh Ni grids (EMS FCF100-Ni). For immunohistochemistry, microetching was done with 10% periodic acid and eluting of osmium with 10% sodium metaperiodate for 15 min at room temperature on parafilm. Grids were rinsed with water three times, followed by 0.5 M glycine quench and then incubated in blocking solution (0.5% BSA, 0.5% ovalbumin in PBST) at room temperature for 20 min. Primary rabbit anti-GFP (1:300; MBL International) was diluted in the same blocking solution and incubated overnight at 4 °C. The next day, grids were rinsed in PBS three times and incubated in secondary antibody (1:10 10 nm gold-conjugated IgG TED Pella 15732) for 1 h at room temperature and rinsed with PBST followed by water. For each staining set, secondary-only staining was simultaneously performed to control for any non-specific binding. Grids were contrast stained for 30 s in 3.5% uranyl acetate in 50% acetone followed by staining in 0.2% lead citrate for 90 s. Samples were imaged using a JEOL JEM-1400 TEM at 120 kV and images were collected using a Gatan Orius digital camera. Adjacent sections from all samples were also tested for ChR2::YFP and YFP signal using fluorescent microscopy to confirm the viral labelling. All image analyses were performed by experimenters blinded to the experimental conditions. Secondary antibody-only controls were also performed to establish immunogold labelling background amounts. Myelinated axons with more than three independent immunogold spheres were counted as positive labelling. The g-ratios were calculated by dividing the shortest axonal diameter by the corresponding axonal-plus-sheath diameter (diameter of axon/diameter of axon plus myelin sheath). Between 296 and 446 axons were analysed per animal. Myelinated axon density was analysed by quantifying the number of myelinated axons at each size per ×10,000 electron micrograph. An average of 45 images were quantified and the total number of myelinated axons was divided by the total area quantified per animal. Group means are calculated on a per mouse basis (not per axon or per image).

### Statistical analysis

The experimenter was blinded to the genotype of the animals during behavioural testing, microscopy imaging and histological analyses. All optogenetic real-time place preference experiments, CPP and CPA pre-test–post-test preference comparisons, and novel-object recognition test data that showed Gaussian distribution were assessed using two-tailed paired t-tests. In cases in which normal distributions were not assumed for CPP preference data, we performed Wilcoxon non-parametric signed-rank tests. For all histological analysis and CPP Z-scores with normal distributions, group mean differences were analysed using unpaired two-tailed t-tests. For data that did not pass the normality test, Mann–Whitney non-parametric tests were used. All statistical analyses were performed using GraphPad Prism Software.

### Reporting summary

Further information on research design is available in the Nature Portfolio Reporting Summary linked to this article.

### Data availability

All data are available in the main text or the Extended Data. The snRNA-seq data are available through GEO (accession number GSE255072). Source data are provided with this paper.

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

**Acknowledgements** We thank A. M. Klawonn for helpful discussions and technical assistance and E. Nestler for valuable insights and feedback. All graphics were created with Biorender. com. Funding was as follows: Gatsby Charitable Foundation (Gatsby Initiative in Brain Development and Psychiatry, to M.M. and R.C.M.), Wu Tsai Neurosciences Institute NeuroChoice Initiative (to R.C.M.), National Institute of Neurological Disorders and Stroke (R01NS092597 to M.M.), NIH Director's Pioneer Award (DP1NS111132 to M.M.), National Institute for Drug Abuse (P50DA042012 to R.C.M. and T32DA035165 and K99DA056573 to M.B.P.), National Cancer Institute (P50CA165962, R01CA258384, U19CA264504 to M.M.), Robert J. Kleberg, Jr and Helen C. Kleberg Foundation (to M.M.), Cancer Grand Challenges (OT2CA278688, CGCATF-2021/100012) and Cancer Research UK (to M.M.), Maternal and Child Health Research Institute at Stanford University Postdoctoral Award (to B.Y.), Dean's Postdoctoral Fellowship at Stanford University (to B.Y.).

**Author contributions** B.Y. and M.M. conceptualized this work. B.Y., M.B.P., R.C.M. and M.M. developed the methodology. B.Y., M.B.P., K.M., I.J.C., K.T., L.N., D.C.-E., K.S., A.E.R. and R.D. carried out the investigation. B.Y. produced the visualization. M.M. and R.C.M. acquired funding. B.Y. was responsible for project administration. M.M. and R.C.M. conducted the supervision. B.Y. and M.M. wrote the original draft. B.Y., M.B.P., R.C.M. and M.M. reviewed and edited the final manuscript.

**Competing interests** R.C.M. is now on leave from Stanford, functioning as the Chief Scientific Officer at Bayshore Global Management. He is on the scientific advisory boards of MapLight Therapeutics, MindMed, Bright Minds Biosciences and Aelis Farma. M.M. holds equity in MapLight Therapeutics and CARGO Biosciences. The remaining authors declare no competing interests.

**Additional information**
**Correspondence and requests for materials** should be addressed to Michelle Monje.

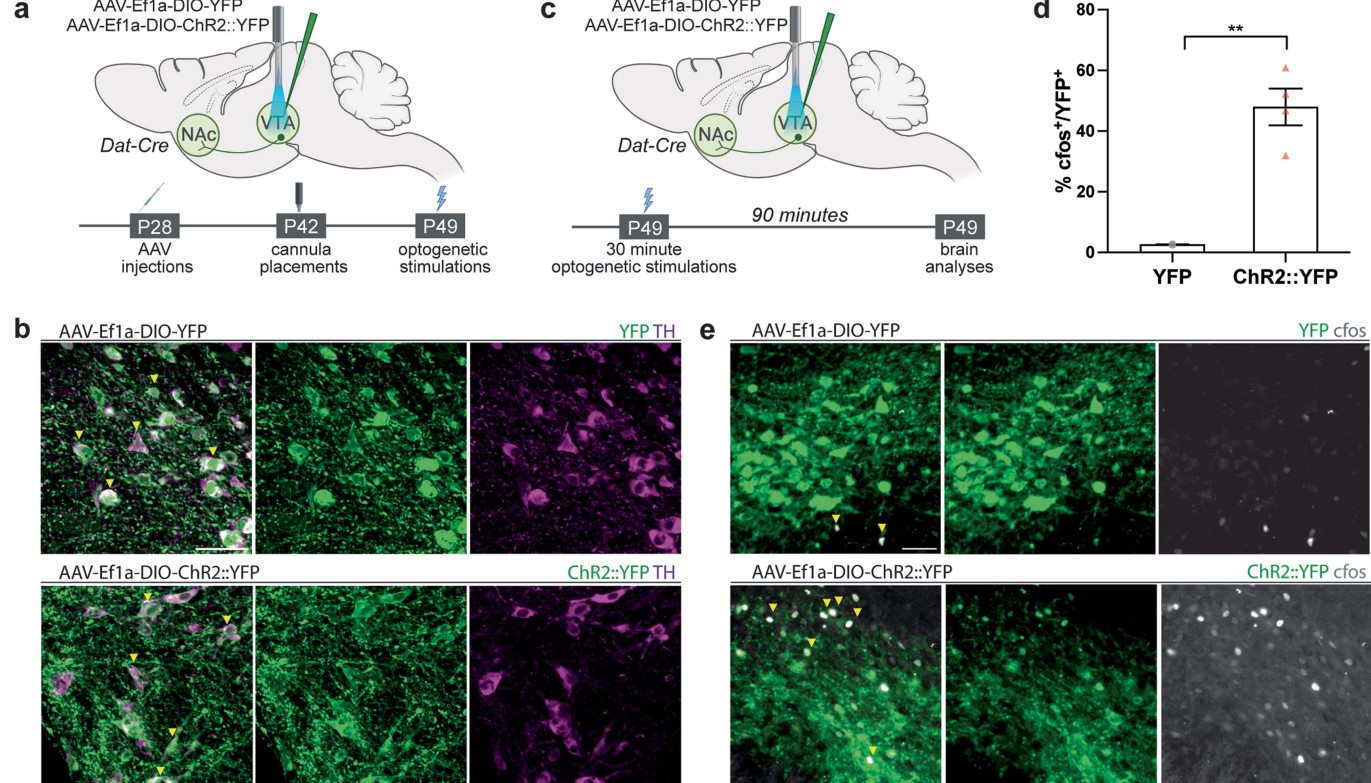

**Extended Data Fig. 1 | Optogenetic activation of dopaminergic neuron activity. a**, Experimental timeline for optogenetics experiments. **b**, Colocalization of YFP (green, panels above) or ChR2::YFP (green, panels below) with the dopaminergic neuron marker, TH (magenta) in VTA, scale bar = 50 μm. **c**, Experimental timeline for optogenetics stimulations. **d**, 30 Hz optogenetic stimulation of ChR2::YFP⁺ neurons increases cfos labeling in VTA

within 90 min (YFP, n = 3; ChR2::YFP, n = 4). **e**, Colocalization of YFP (green, panels above) or ChR2::YFP (green, panels below) with the immediate early gene marker, cfos (grey) indicating neuronal activity in VTA. Unpaired two-tailed t-test, **p < 0.01. Each data point represents a mouse, data shown as mean, error bars indicate SEM. **a,c**, Schematics created with BioRender.com.

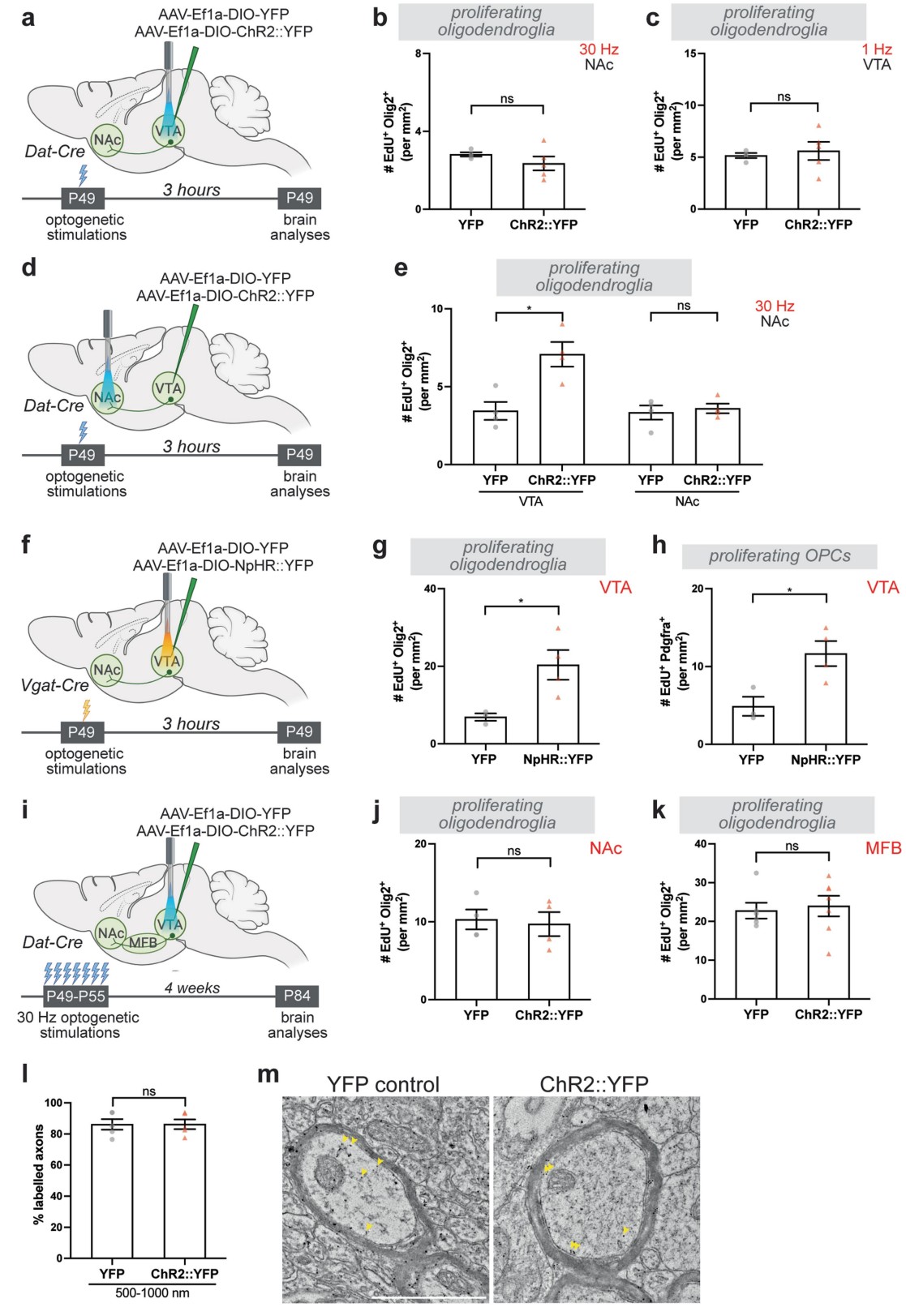

**Extended Data Fig. 2** | See next page for caption.

**Extended Data Fig. 2 | Dopaminergic neuron activity promotes new oligodendroglia in VTA. a**, Experimental paradigm for acute dopaminergic neuron stimulation in VTA; 30 min optogenetic stimulation followed by brain analysis after 3 h. **b**, 30 Hz VTA stimulation of dopaminergic neurons does not change proliferating oligodendroglia (Olig2$^+$ EdU$^+$) in NAc (YFP, n = 4 mice; ChR2::YFP, n = 5 mice). **c**, 1 Hz VTA stimulation of dopaminergic neurons does not change proliferating oligodendroglia (Olig2$^+$ EdU$^+$) in VTA (YFP, n = 4 mice; ChR2::YFP, n = 5 mice). **d**, Experimental paradigm for acute dopaminergic neuron stimulation in NAc; 30 min optogenetic stimulation followed by brain analysis after 3 h. **e**, 30 Hz stimulation of dopaminergic neurons increases proliferating oligodendroglia (Olig2$^+$ EdU$^+$) in VTA but not in NAc (YFP, n = 4 mice; ChR2::YFP, n = 4 mice). **f**, Experimental paradigm for acute GABAergic neuron inhibition in VTA; 30-min optogenetic inhibition followed by brain analysis after 3 h. **g**, VTA inhibition of GABAergic neurons increases proliferating oligodendroglia (Olig2$^+$ EdU$^+$) and **h**, proliferating OPCs (Pdgfrα$^+$ EdU$^+$) in VTA (YFP, n = 3 mice; NpHR::YFP, n = 4 mice). **i**, Experimental paradigm for chronic dopaminergic neuron stimulation in VTA; 30 Hz optogenetic stimulation for 10 min a day for 7 days, followed by brain analysis 4 weeks after this paradigm. Chronic 30 Hz dopaminergic neuron stimulation does not change in **j**, NAc (YFP, n = 4 mice; ChR2::YFP, n = 4 mice) or **k**, MFB (YFP, n = 6 mice; ChR2::YFP, n = 7 mice). **l**, Percentage of immunogold labelled medium sized (500-1000 nm) axons in VTA (YFP, n = 5 mice; ChR2::YFP, n = 5 mice). **m**, electron micrographs of immunogold labelled myelinated axons, scale bar=1 µm. Unpaired two-tailed t-test, ns (not significant) p > 0.5, *p < 0.05. Each data point represents a mouse, data shown as mean, error bars indicate SEM. **a**,**d**,**f**,**i**, Schematics created with BioRender.com.

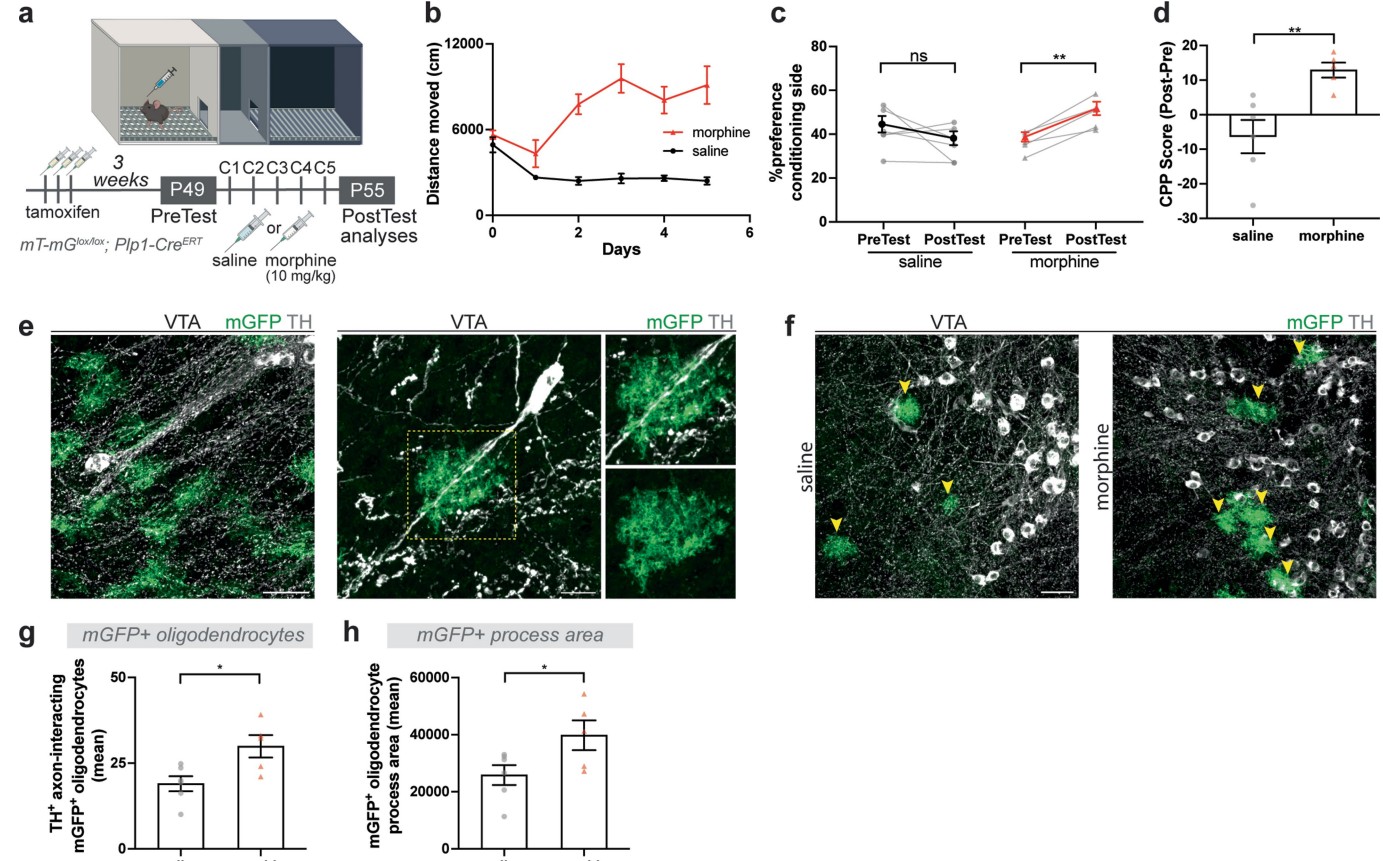

**Extended Data Fig. 3 | Morphine increases oligodendrocytes in VTA.**
**a**, Experimental paradigm, *mT-mG^{lox/lox}; Plp1-Cre^{ERT}* mice are administered tamoxifen 3 weeks prior to morphine CPP. Mice groups injected with saline and EdU or morphine (10 mg/kg) and EdU at 7 weeks and brains analyzed after post-test. **b**, Morphine conditioned mice exhibit increased locomotor sensitivity (saline, n = 6 mice; morphine, n = 5 mice). **c**, Morphine injected mice acquire a place preference for the morphine conditioning chamber, whereas saline mice do not show a strong preference. Graph shows the percentage of time spent in conditioning chamber, comparing pre-test and post-test (saline, n = 6 mice; morphine, n = 5 mice). **d**, Morphine group show increased CPP Score (post-test

− pre-test preference) compared to saline controls. **e**, mGFP⁺ oligodendrocytes in VTA after morphine CPP, dotted lines focus on an oligodendrocyte interacting with the proximal section of a dopaminergic axon. **e**, mGFP⁺ oligodendrocytes in VTA. TH marks DA neurons (grey), mGFP⁺ oligodendrocytes (green), scale bars= 50 µm. **g**, morphine increases number of mGFP⁺ oligodendrocytes interacting with TH⁺ dopaminergic axons and **h**, area of mGFP⁺ oligodendrocyte processes in VTA. **c**, paired two-tailed t-test, **d**, **g**, **h**, unpaired two-tailed t-test, ns (not significant) p > 0.5, *p < 0.05, **p < 0.01. Each data point represents a mouse, data shown as mean, error bars indicate SEM. **a**, Schematic created with BioRender.com.

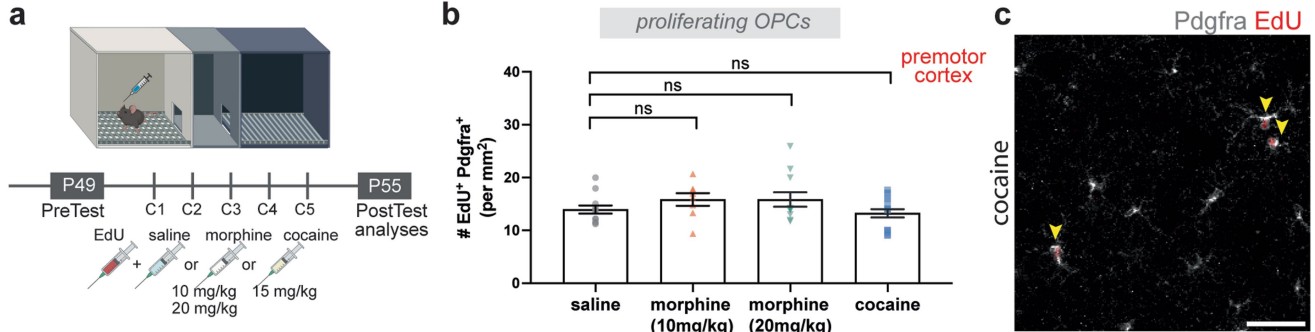

**a** P49 PreTest — EdU — C1 C2 C3 C4 C5 — P55 PostTest analyses
saline, morphine 10 mg/kg 20 mg/kg, cocaine 15 mg/kg

**b** proliferating OPCs

**c** Pdgfra EdU — cocaine — premotor cortex

**Extended Data Fig. 4 | Morphine or cocaine show no effect on cortical OPC proliferation. a**, Experimental paradigm, mice groups injected with saline and EdU, or morphine (10 mg/kg, or 20 mg/kg), or cocaine (15 mg/kg) and EdU at 7 weeks and brains analyzed after post-test. **b**, Neither morphine nor cocaine changes OPC proliferation in premotor cortex (saline, n = 12 mice; morphine-10 mg/kg, n = 8 mice; morphine-20 mg/kg, n = 11 mice; cocaine, n = 16 mice).

**c**, Proliferating OPCs in premotor cortex after cocaine CPP, arrowheads denote proliferating OPCs (Pdgfrα⁺ EdU⁺), Pdgfrα⁺ (grey), EdU⁺ (red). ns = not significant, p > 0.5; unpaired two-tailed t-test. Each data point represents a mouse, data shown as mean, error bars indicate SEM. **a**, Schematic created with BioRender.com.

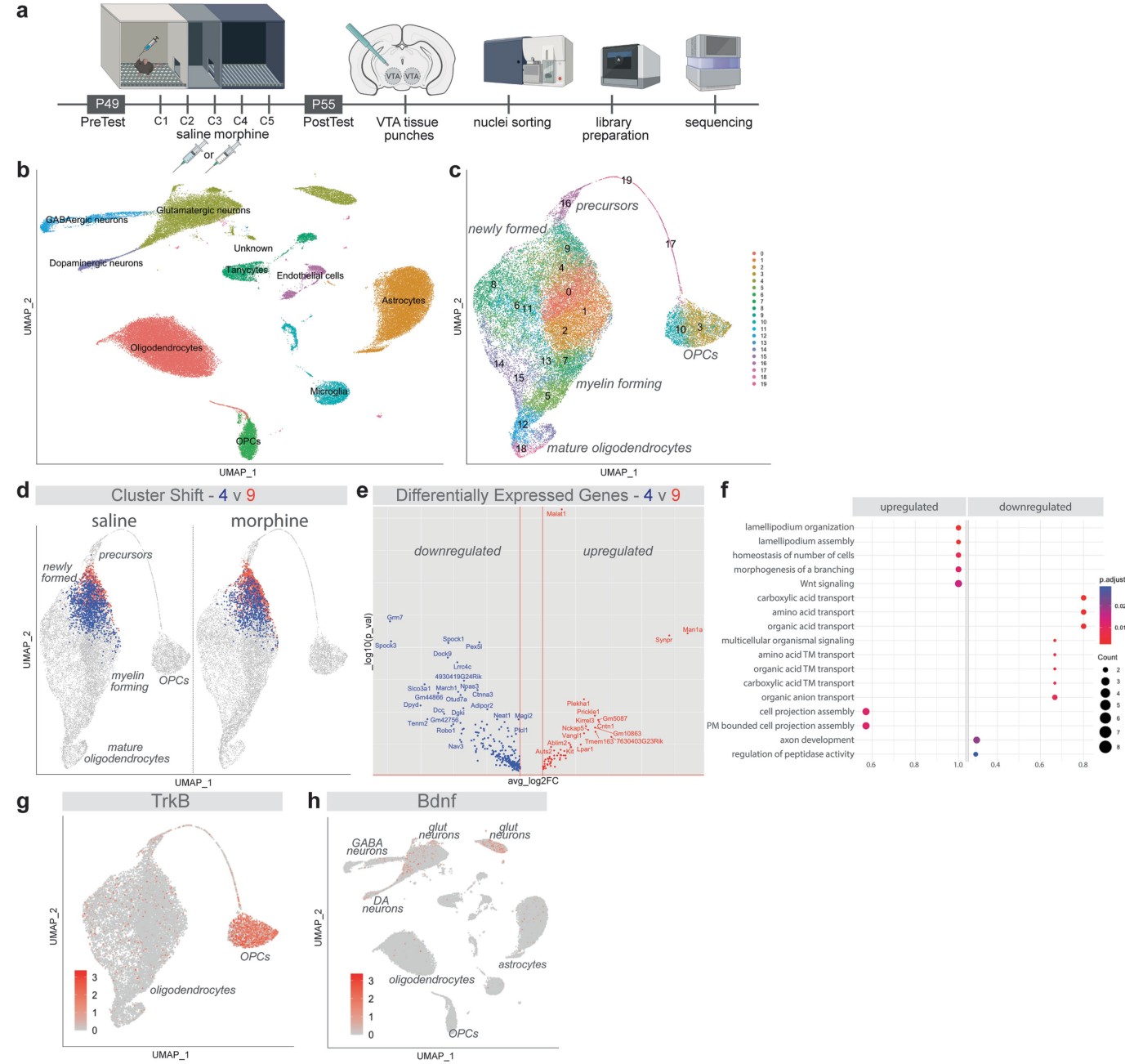

**Extended Data Fig. 5 | Morphine shifts oligodendroglial gene expression program towards differentiation. a**, Experimental paradigm, mice groups injected with saline or morphine (10 mg/kg) at 7 weeks and after post-test, VTA tissue punches are collected and nuclei sorted, then libraries are prepared to perform snRNA-seq. **b**, UMAP of all cell types in VTA by snRNA-seq. snRNA-seq identifies different cellular populations in VTA, GABAergic neurons, glutamatergic neurons, dopaminergic neurons, tanycytes, endothelial cells, astrocytes, oligodendrocytes, OPCs, astrocytes and microglia. **c**, UMAP of oligodendroglia in VTA by snRNA-seq. snRNA-seq identifies different 19 different subpopulations oligodendroglia at various stages of maturity in VTA.

**d**, Morphine causes shifts in oligodendroglial subpopulation clusters. For example, cluster 4 shifts towards cluster 9 with morphine exposure. **e**, Differentially expressed genes defines the changes between cluster 4 to cluster 9 shift. **f**, Morphine causes shifts in oligodendroglial transcription which is consistent with a more differentiated cell stage (TM: transmembrane; PM: plasma membrane). **g**, TrkB (*Ntrk2* gene) is highly expressed by OPCs in the VTA and downregulated as they differentiate into oligodendrocytes. **h**, *Bdnf* is expressed by neurons including dopaminergic neurons in VTA. **a**, Schematic created with BioRender.com.

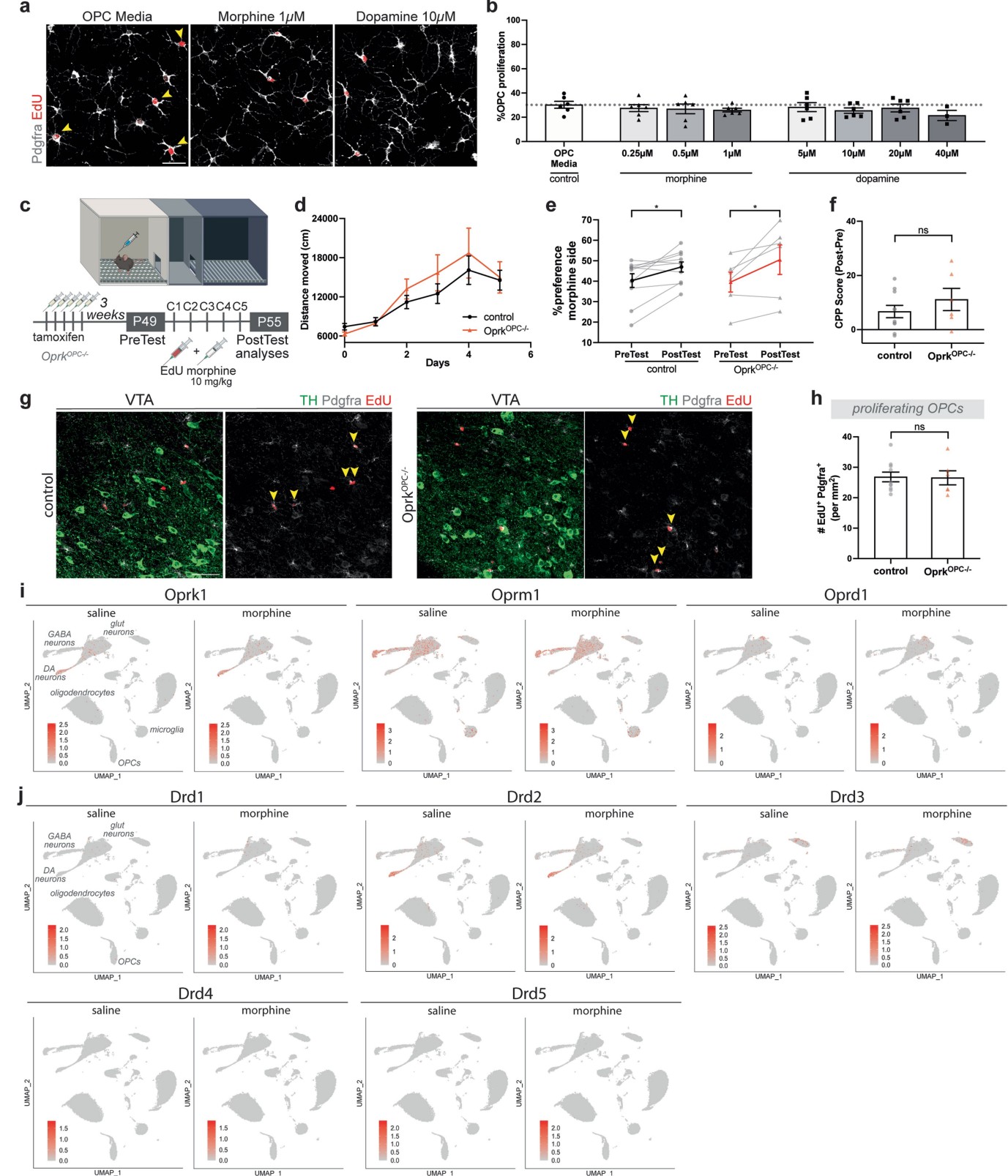

**Extended Data Fig. 6** | See next page for caption.

**Extended Data Fig. 6 | Morphine exerts no effect on OPC proliferation in vitro or through Oprk. a**, Proliferating OPCs (Pdgfrα⁺ EdU⁺) in vitro when exposed to control OPC media, 1 μM morphine, or 10 μM dopamine. **b**, Quantification of proliferating OPCs (Pdgfrα⁺ EdU⁺) after exposure to different concentrations of morphine and dopamine show similar OPC proliferation compared to OPC media control (each point indicates one technical replicate). **c**, Experimental paradigm, tamoxifen was administered at 4 weeks of age to conditionally knockout *Oprk1* from OPCs, 3 weeks prior to start of the morphine CPP. During conditioning days mice groups injected with morphine (10 mg/kg) and EdU at 7 weeks and brains analyzed after post-test. **d**, Morphine increases locomotor sensitivity in both control and *Oprk*^OPC–/–. **e**, Both control and *Oprk*^OPC–/– mice acquire a place preference for the morphine conditioning chamber and **f**, Show similar CPP score. **g**, Proliferating OPCs in VTA after morphine CPP, arrowheads denote proliferating OPCs (Pdgfrα⁺ EdU⁺), TH marks dopaminergic neurons (green), Pdgfrα⁺ (grey), EdU⁺ (red), scale bar= 50 μm. **h**, Control and *Oprk*^OPC–/– mice show similar OPC proliferation in response to morphine CPP. **i-j**, UMAP of single nucleus sequencing (sn-Seq) in VTA. Cell cluster identity is labeled in the first panel. **i**, Oligodendroglial cells in the VTA do not express opioid receptors (*Oprk1, Oprm1, Oprd1*) and this does not change after morphine CPP. **j**, Oligodendroglial cells in the VTA do not express dopamine receptors (*Drd1-5*) and this does not change after morphine CPP. **d, e, f, h**, (control, n = 10 mice; *Oprk*^OPC–/–, n = 6 mice) **d**, paired two-tailed t-test, **b, f, h**, unpaired two-tailed t-test, ns (not significant) p > 0.5, *p < 0.05. Each data point represents a mouse, data shown as mean, error bars indicate SEM. **c**, Schematic created with BioRender.com.

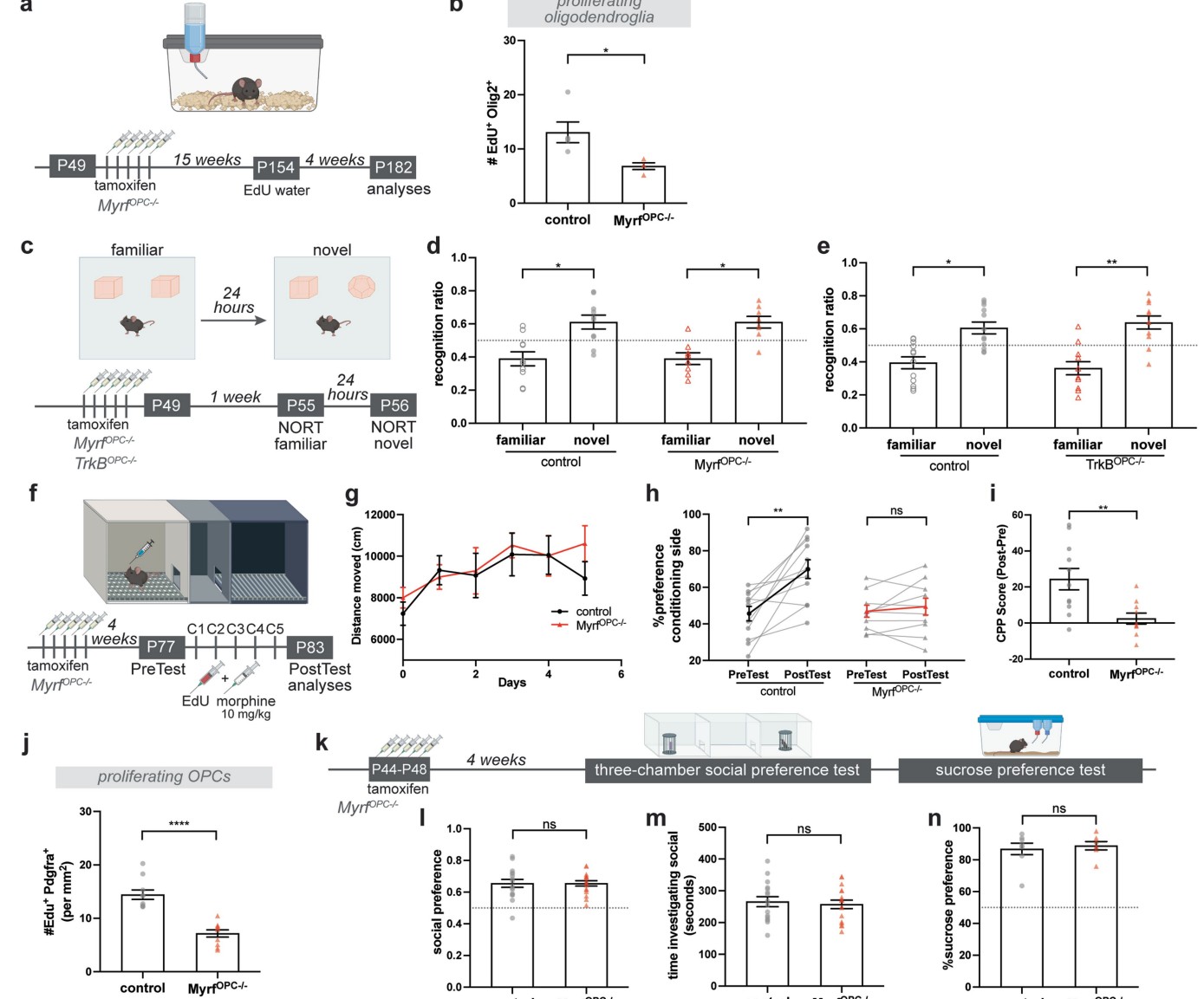

**Extended Data Fig. 7 | *Myrf* loss causes reduced oligodendrogenesis but not memory impairment or anhedonia at the time points tested. a**, Experimental paradigm for determining the effects of *Myrf* loss on oligodendrogenesis; 15 weeks after conditional knock-out of *Myrf* by tamoxifen administration in *Myrf*^OPC−/− (*Myrf*^lox/lox;*Pdgfrα-Cre*^ERT) and littermate controls, EdU is added to drinking water to measure the baseline levels of oligodendrogenesis within 4 weeks. **b**, *Myrf*^OPC−/− animals show reduced proliferating oligodendroglia (Pdgfrα⁺ Olig⁺) in corpus callosum white matter compared to controls (control, n = 5 mice; *Myrf*^OPC−/−, n = 4 mice). **c**, Experimental paradigm for novel object recognition test; mice are introduced to two identical objects and tested 24 h after for memory by switch one of the objects with a novel object. 1 week after the end of tamoxifen administration *Myrf*^OPC−/− (*Myrf*^lox/lox;*Pdgfrα-Cre*^ERT) or *TrkB*^OPC−/− (*TrkB*^lox/lox;*Pdgfrα-Cre*^ERT) animals and littermate controls are tested in novel object recognition test. This timeline is identical to CPP paradigms tested in Fig. 4a and g. **d**, Both control and *Myrf*^OPC−/− animals spend more time with novel object compared to the familiar object (control, n = 10 mice; *Myrf*^OPC−/−, n = 8 mice). **e**, Both control and *TrkB*^OPC−/− animals spend more time with novel object compared to the familiar object (control, n = 12 mice; *Myrf*^OPC−/−, n = 11 mice). **f**, Experimental paradigm for *Myrf*^OPC−/− and littermate controls; conditional knock-out performed with tamoxifen administration at 7-weeks of

age, 4-weeks before CPP. Mice groups injected with morphine (10 mg/kg) and EdU during conditioning sessions and brains analyzed after post-test. **g**, Both control and *Myrf*^OPC−/− mice exhibits increased locomotor sensitivity during morphine conditioning (control, n = 11 mice; *Myrf*^OPC−/−, n = 10 mice). **h**, Control mice acquire a place preference for the morphine conditioning chamber, whereas *Myrf*^OPC−/− mice do not (control, n = 11 mice; *Myrf*^OPC−/−, n = 10 mice). **i**, *Myrf*^OPC−/− animals show decreased CPP Score (post-test − pre-test preference) compared to controls. **j**, Loss of Myrf decreases number of proliferating OPCs (Pdgfrα⁺ EdU⁺) in VTA after morphine CPP (control and *Myrf*^OPC−/−, n = 10 mice). **k**, Experimental paradigm for *Myrf*^OPC−/− and littermate controls; conditional knock-out performed with tamoxifen administration at 7-weeks of age, 4-weeks prior to behavioral tests. **l**, Both control and *Myrf*^OPC−/− mice show strong preference for the social subject during a three-chamber social preference test and **m**, spent similar time investigating social subject (control, n = 16 mice; *Myrf*^OPC−/−, n = 17 mice). **n**, Both control and *Myrf*^OPC−/− mice show strong preference for sucrose over water (control, n = 8 mice; *Myrf*^OPC−/−, n = 7 mice). **b, i, j, l, m, n**, unpaired two-tailed t-test, **d, e, h**, paired two-tailed t-test for each genotype, ns (not significant) p > 0.5, *p < 0.05, **p < 0.01, ***p < 0.001, ****p < 0.0001. Each data point represents a mouse, data shown as mean, error bars indicate SEM. **a,c,f,k**, Schematics created with BioRender.com.

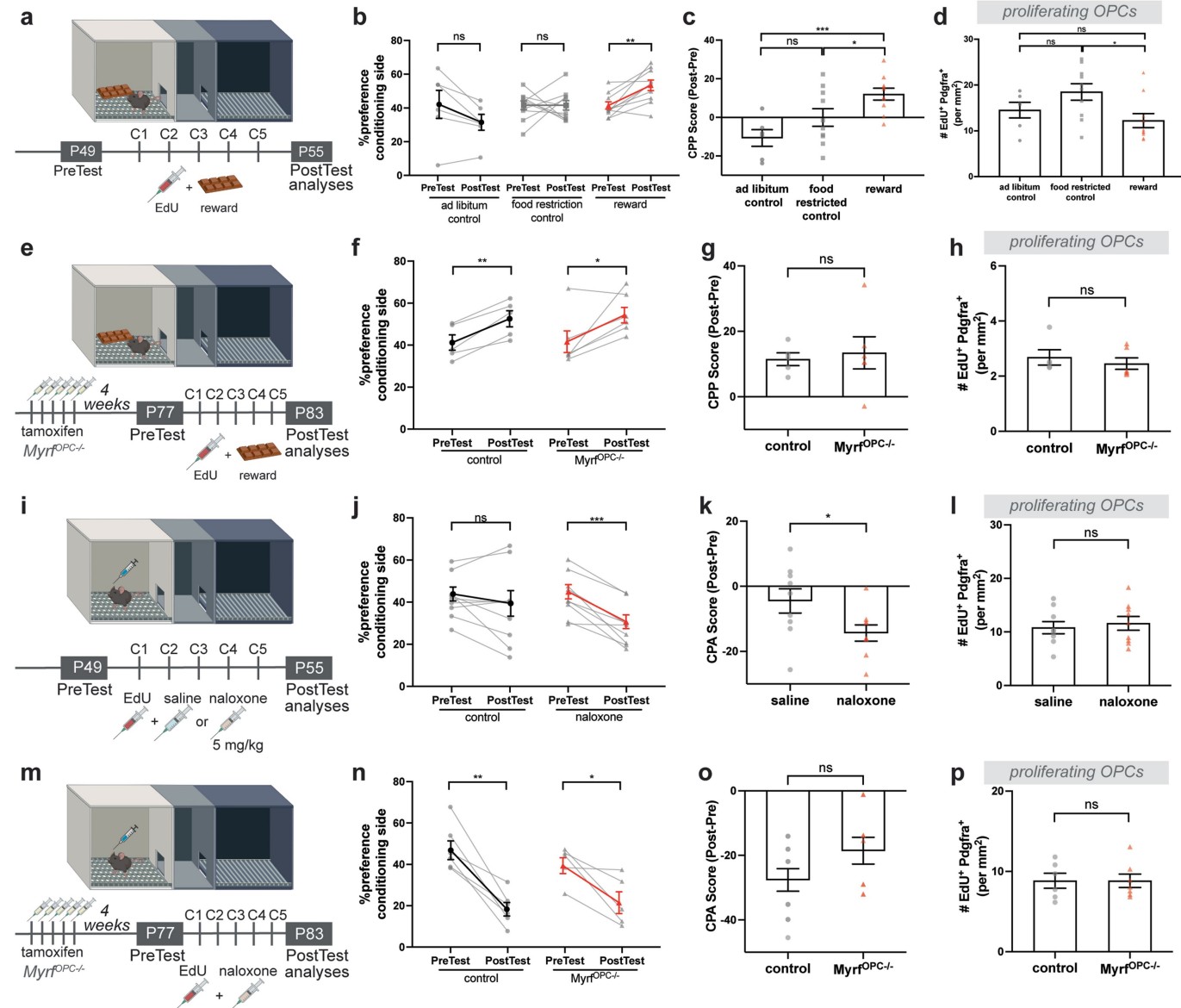

**Extended Data Fig. 8 | Neither food reward nor naloxone aversion requires oligodendrogenesis. a**, Experimental paradigm for food-induced CPP; mice were conditioned with chocolate pellet (after food restriction), one control group fed ad libitum while another control group is food restricted. All animals received EdU injections during conditioning sessions for tracing proliferating cells. **b**, Food reward conditioned animals exhibit a strong preference to the reward-paired chamber, whereas control groups do not (ad libitum control, n = 6 mice; food restricted control, n = 10 mice; food reward, n = 10 mice). **c**, Food reward conditioned animals show increased CPP Score (post-test – pre-test preference) compared to the control groups. **d**, Food reward conditioning does not change the number of proliferating OPCs (Pdgfrα⁺ EdU⁺) in VTA. Food restricted control group shows a slight increase of proliferating OPCs compared to the food reward group due to the known effects of calorie restriction on oligodendroglia dynamics (ad libitum control, n = 6 mice; food restricted control, n = 10 mice; food reward, n = 10 mice). **e**, Experimental paradigm for *Myrf^{OPC–/–}* (*Myr^{lox/lox}; Pdgfr*α-*Cre^{ERT}*) and littermate controls; conditional knock-out performed with tamoxifen administration at 7-weeks of age, 4-weeks before CPP. Mice groups are conditioned with chocolate pellet food reward and administered with EdU during conditioning sessions and brains analyzed immediately after post-test. **f**, Both control and *Myrf^{OPC–/–}* mice acquire a place preference for the food reward-paired chamber (control, n = 5 mice; *Myrf^{OPC–/–}*, n = 6 mice). **g**, *Myrf^{OPC–/–}* animals similar CPP Score (post-test – pre-test preference) to controls. **h**, Loss of Myrf does not change the number of proliferating OPCs

(Pdgfrα⁺ EdU⁺) in VTA after food reward CPP (control, n = 5 mice; *Myrf^{OPC–/–}*, n = 6 mice). **i**, Experimental paradigm for the opioid receptor antagonist, naloxone, conditioning; mice were conditioned with naloxone (5 mg/kg) or saline and injected with EdU during conditioning sessions. **j**, Naloxone conditioned animals exhibit a strong avoidance to the naloxone-paired chamber, whereas saline animals do not show a preference (saline and naloxone, n = 9 mice). **k**, Naloxone conditioned animals show decreased CPP Score compared to the saline groups. **l**, Naloxone conditioned place avoidance does not change the number of proliferating OPCs (Pdgfrα⁺ EdU⁺) in VTA (saline and naloxone, n = 9 mice). **m**, Experimental paradigm for *Myrf^{OPC–/–}* (*Myrf^{lox/lox}; Pdgfr*α-*Cre^{ERT}*) and littermate controls; conditional knock-out performed with tamoxifen administration at 7-weeks of age, 4-weeks before naloxone conditioning. Mice groups are conditioned with naloxone and administered with EdU during conditioning sessions and brains analyzed after post-test. **n**, Both control and *Myrf^{OPC–/–}* mice acquire a place avoidance for the naloxone-paired chamber (control, n = 6 mice; *Myrf^{OPC–/–}*, n = 5 mice). **o**, *Myrf^{OPC–/–}* animals similar CPA to controls. **p**, Loss of Myrf does not change number of proliferating OPCs (Pdgfrα⁺ EdU⁺) in VTA after naloxone CPA (control, n = 6 mice; *Myrf^{OPC–/–}*, n = 7 mice). **c, d, g, h, k, l, o, p**, unpaired two-tailed t-test, **b, f, j, n**, paired two-tailed t-test, ns (not significant) p > 0.5, *p < 0.05, **p < 0.01, ***p < 0.001. Each data point represents a mouse, data shown as mean, error bars indicate SEM. **a, e, i, m**, Schematics created with BioRender.com.

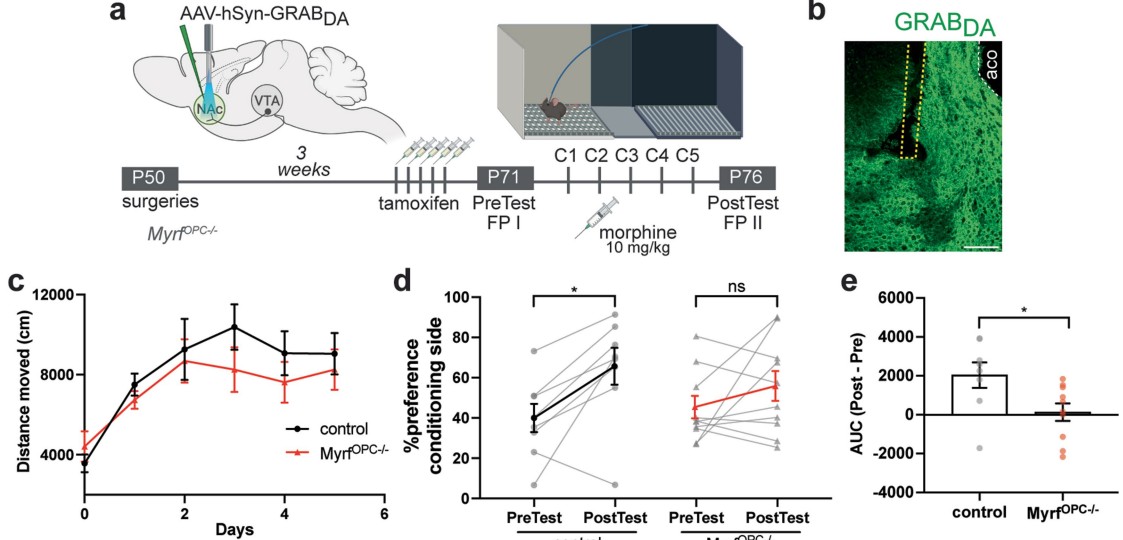

**Extended Data Fig. 9 | Morphine-induced oligodendrogenesis alters dopamine dynamics in NAc. a**, Experimental paradigm to determine the effects of morphine on dopamine dynamics in *Myrf*^OPC−/−^ (*Myrf*^lox/lox^; *Pdgfr*α*-Cre*^ERT^) and littermate controls; conditional knock-out performed with tamoxifen administration prior to CPP. Dopamine dynamics measured with fiber photometry during pre-test and post-test as mice freely explored CPP chambers. Mice groups injected with morphine (10 mg/kg) during conditioning sessions. **b**, Fiber photometry cannula positioning in NAc outlined by yellow dashed lines, white dashed line marks anterior commissure (aco). GRAB_DA (green)

expression in NAc below the cannula, scale bar=100 μm. **c**, Morphine increases locomotor sensitization in both control and *Myrf*^OPC−/−^ mice. **d**, Control mice acquire a place preference for the morphine conditioning chamber, whereas *Myrf*^OPC−/−^ mice do not (control, n = 8 mice; *Myrf*^OPC−/−^, n = 10 mice). **e**, Control mice release more dopamine in post-test compared to *Myrf*^OPC−/−^ mice (control, n = 8 mice; *Myrf*^OPC−/−^, n = 10 mice). AUC (area under the curve), **d**, Wilcoxon matched-pairs test, **f, g, h**, unpaired two-tailed t-test, ns (not significant) p > 0.5, *p < 0.05. Each data point represents a mouse, data shown as mean, error bars indicate SEM. **a**, Schematic created with BioRender.com.

# Reporting Summary

## Statistics

For all statistical analyses, confirm that the following items are present in the figure legend, table legend, main text, or Methods section.

| n/a | Confirmed | |
|---|---|---|
| ☐ | ☒ | The exact sample size (*n*) for each experimental group/condition, given as a discrete number and unit of measurement |
| ☐ | ☒ | A statement on whether measurements were taken from distinct samples or whether the same sample was measured repeatedly |
| ☐ | ☒ | The statistical test(s) used AND whether they are one- or two-sided<br>*Only common tests should be described solely by name; describe more complex techniques in the Methods section.* |
| ☐ | ☒ | A description of all covariates tested |
| ☐ | ☒ | A description of any assumptions or corrections, such as tests of normality and adjustment for multiple comparisons |
| ☐ | ☒ | A full description of the statistical parameters including central tendency (e.g. means) or other basic estimates (e.g. regression coefficient) AND variation (e.g. standard deviation) or associated estimates of uncertainty (e.g. confidence intervals) |
| ☒ | ☐ | For null hypothesis testing, the test statistic (e.g. *F*, *t*, *r*) with confidence intervals, effect sizes, degrees of freedom and *P* value noted<br>*Give P values as exact values whenever suitable.* |
| ☒ | ☐ | For Bayesian analysis, information on the choice of priors and Markov chain Monte Carlo settings |
| ☒ | ☐ | For hierarchical and complex designs, identification of the appropriate level for tests and full reporting of outcomes |
| ☒ | ☐ | Estimates of effect sizes (e.g. Cohen's *d*, Pearson's *r*), indicating how they were calculated |

*Our web collection on statistics for biologists contains articles on many of the points above.*

## Software and code

Policy information about availability of computer code

| Data collection | All behavioral data were recoded as videos. |
|---|---|
| Data analysis | Behavioral experiments were analyzed using Ethovision XT software (Noldus) in an automated and condition-blinded manner. Fiji was used for confocal and electron microscopy analysis. MBF Stereoinvestigator is used for analysis of fluorescence stereomicropscope images. All statistical analysis were performed using Graphad Prism Software. Fiber photometry signal processing was performed in MATLAB (MathWorks). |

For manuscripts utilizing custom algorithms or software that are central to the research but not yet described in published literature, software must be made available to editors and reviewers. We strongly encourage code deposition in a community repository (e.g. GitHub). See the Nature Portfolio guidelines for submitting code & software for further information.

## Data

Policy information about availability of data

All manuscripts must include a data availability statement. This statement should provide the following information, where applicable:
- Accession codes, unique identifiers, or web links for publicly available datasets
- A description of any restrictions on data availability
- For clinical datasets or third party data, please ensure that the statement adheres to our policy

All data are available in the main text or the supplementary materials. Sourced data will be available with the final version of the manuscript.

# Human research participants

Policy information about <u>studies involving human research participants and Sex and Gender in Research.</u>

| | |
|---|---|
| Reporting on sex and gender | *Use the terms sex (biological attribute) and gender (shaped by social and cultural circumstances) carefully in order to avoid confusing both terms. Indicate if findings apply to only one sex or gender; describe whether sex and gender were considered in study design whether sex and/or gender was determined based on self-reporting or assigned and methods used. Provide in the source data disaggregated sex and gender data where this information has been collected, and consent has been obtained for sharing of individual-level data; provide overall numbers in this Reporting Summary. Please state if this information has not been collected. Report sex- and gender-based analyses where performed, justify reasons for lack of sex- and gender-based analysis.* |
| Population characteristics | *Describe the covariate-relevant population characteristics of the human research participants (e.g. age, genotypic information, past and current diagnosis and treatment categories). If you filled out the behavioural & social sciences study design questions and have nothing to add here, write "See above."* |
| Recruitment | *Describe how participants were recruited. Outline any potential self-selection bias or other biases that may be present and how these are likely to impact results.* |
| Ethics oversight | *Identify the organization(s) that approved the study protocol.* |

Note that full information on the approval of the study protocol must also be provided in the manuscript.

# Field-specific reporting

Please select the one below that is the best fit for your research. If you are not sure, read the appropriate sections before making your selection.

☒ Life sciences    ☐ Behavioural & social sciences    ☐ Ecological, evolutionary & environmental sciences

For a reference copy of the document with all sections, see nature.com/documents/nr-reporting-summary-flat.pdf

# Life sciences study design

All studies must disclose on these points even when the disclosure is negative.

| | |
|---|---|
| Sample size | . Sample sizes were guided by power calculation and based on previous publications (Gibson et al 2014, Science)  Sample sizes of 3–20 animals were sufficient to determine significance both in histological analysis and behaviour tests. |
| Data exclusions | For CPP tests, mice that show strong preference (>75%) to either chamber during PreTest was excluded from further analysis. For novel object recognition, animals that explored the objects less than 20 seconds were excluded from the analysis. Brain tissue that is damaged during perfusion or tissue processing, or with inadequate viral labeling were excluded from histological analysis. |
| Replication | Every experiment were repeated multiple times to replicate in independent cohorts of animals, and all attempts at replication was successful. Reported findings were reproduced across animals in histological, and behavioral experiments. Total number of animals were reported in each figure, and group means were used for statistical analysis as indicated in the figure legends, and methods section. |
| Randomization | Animals were randomized by cage prior to surgeries. For example, mice were randomly assigned to YFP or ChR2 groups, or wild type animals were randomized for receiving saline, morphine, or cocaine injections. |
| Blinding | All behavioral and histological analysis were conducted blind to genotype, manipulation or to drug treatment. |

# Reporting for specific materials, systems and methods

We require information from authors about some types of materials, experimental systems and methods used in many studies. Here, indicate whether each material, system or method listed is relevant to your study. If you are not sure if a list item applies to your research, read the appropriate section before selecting a response.

## Materials & experimental systems

| n/a | Involved in the study |
|---|---|
| ☐ | ☒ Antibodies |
| ☒ | ☐ Eukaryotic cell lines |
| ☒ | ☐ Palaeontology and archaeology |
| ☐ | ☒ Animals and other organisms |
| ☒ | ☐ Clinical data |
| ☒ | ☐ Dual use research of concern |

## Methods

| n/a | Involved in the study |
|---|---|
| ☒ | ☐ ChIP-seq |
| ☒ | ☐ Flow cytometry |
| ☒ | ☐ MRI-based neuroimaging |

## Antibodies

| | |
|---|---|
| Antibodies used | Primary antibodies: Goat anti- Pdgfra (1:500; R&D Systems, AF1062), rabbit anti-Olig2 (1:500; Abcam, 7349), rabbit anti-ASPA (1:250; EMD Millipore, ABN1698), rat anti-MBP (1:200; Abcam ab7349), rabbit anti-tyrosine hydroxylase (1:500; Millipore Sigma, AB152), mouse anti-tyrosine hydroxylase (1:250; Novus Biologicals, MAB7566), chicken anti-GFP (1:1000; Aves Labs, GFP-1020), chicken anti-mCherry (1:1000; ab205402), or rabbit anti-cfos (1:500; Santa Cruz Biotechnology, sc-52). secondary antibodies: used at 1:500 concentration including Alexa 488 anti-rabbit (Jackson ImmunoResearch; 711-545-152), Alexa 488 anti-mouse (Jackson ImmunoResearch; 715-545-150), Alexa 488 anti-chicken (Jackson ImmunoResearch; 703-545-155), Alexa 594 anti-chicken (Jackson ImmunoResearch; 703-585-155), Alexa 647 donkey anti-goat (Jackson ImmunoResearch; 705-605-147), Alexa 647 anti-rat (Jackson ImmunoResearch; 712-605-150), Alexa 647 anti-rabbit (Jackson ImmunoResearch; 711-605-152), Alexa 647 donkey anti-goat (Jackson ImmunoResearch; 705-605-147). All used at 1:500 |
| Validation | Immunostaining results were compared with published data for known distribution of labeled structures. |

## Animals and other research organisms

Policy information about studies involving animals; ARRIVE guidelines recommended for reporting animal research, and Sex and Gender in Research

| | |
|---|---|
| Laboratory animals | Hemizygous Dat-Cre (The Jackson Laboratory, 006660) mice were bred with C57BL/6J (The Jackson Laboratory, 000664). Homozygous for Vgat-Cre (The Jackson Laboratory, 028862) mice bred with C57BL/6J (The Jackson Laboratory, 000664). For conditional deletion of Myrf, hemizygous Pdgfra-CreERTM (The Jackson Laboratory, 018280) mice were bred with homozygous Myrflox/lox (The Jackson Laboratory, 010607). For conditional deletion of TrkB, hemizygous Pdgfra-CreERTM (The Jackson Laboratory, 018280) mice were bred with homozygous TrkBlox/lox (MMRC, 033048-UCD). For conditional deletion of Oprk, hemizygous Pdgfra-CreERTM (The Jackson Laboratory, 018280) mice were bred with homozygous Oprk1lox/lox (The Jackson Laboratory, 030076). For oligodendrocyte specific expression of mGFP, homozygous mT/mGlox/lox (The Jackson Laboratory, 007676) animals were crossed with Plp1-CreERT (The Jackson Laboratory, 005975). For testing morphine effects on oligodendroglial cells, wild type C57BL/6J (The Jackson Laboratory, 000664) mice were used. All animals for housed in a 12-hour light/dark cycle with unrestricted access to food and water. Behavioral experiments were performed during the same circadian period (7:00-19:00). |
| Wild animals | No wild animals were used. |
| Reporting on sex | Both male and female mice were used equally in all experiments. |
| Field-collected samples | No Samples were field-collected. |
| Ethics oversight | All procedures were performed in accordance with guidelines set in place by National Institutes of Health and approved by the Stanford University Institutional Care and Use Committee (IACUC). |

Note that full information on the approval of the study protocol must also be provided in the manuscript.

