## [Peer Review file · Nature]

Manuscript Title: Myelin plasticity in ventral tegmental area is required for opioid reward

Reviewer Comments & Author Rebuttals

Reviewer Reports on the Initial Version:

Referees' comments:

Referee #1 (Remarks to the Author):

"Myelin plasticity in ventral tegmental area is required for opioid reward" presents data that activation of the Dopaminergic neuron reward circuit stimulates the oligodendrocyte lineage and that this plays a role in regulating opioid-driven reward-seeking behaviour. This is a very interesting fundamental finding, and is another piece of evidence that places active regulation of the oligodendrocyte lineage as a critical aspect of neural circuit function, and in this case, arguably dysfunction. The study takes an ontogenetic-based approach in rodents to stimulate dopaminergic neuron activity in the VTA region of the brainstem and finds, quite convincingly that oligodendrocyte progenitor cells (OPCs) are stimulated to proliferate, mirroring similar observations by the same group and others in different brain areas. The study also shows that this effect is quite local, regulating OPCs in the VTA region, but not in target brain regions, e.g the NAc. Longer term ontogenetic stimulation results in local (VTA) increased oligodendrogenesis, and myelination, even though the effect sizes are somewhat muted. Very interestingly, application of either morphine or cocaine mirrors (where directly compared, see below) the effects of dopaminergic neuron stimulation on VTA oligodendrocyte lineage behaviour, suggesting the possibility that activity-driven regulation of OPCs and oligodendrocytes might play a functional role in regulating the rear circuit. This premise is supported by evidence gathered using a transgenic mouse in which the generation of new adult oligodendrocytes is impaired, such that the morphine induced changes in behaviour are muted in the mutant animal. As a first step towards investigating the potential circuit-based mechanism by which the oligodendrocyte lineage may influence behaviour, the study also shows that dopamine release in the NAc is reduced in the transgenic animals with impaired oligodendrogenesis. To me, this study has the potential to be of enormous general interest. The observations are novel, the topic of importance and the data as presented so far appear robust, although I will note a lack of expertise in rodent behavioural analyses that mute my own ability to speak to the details thereof. Nonetheless, the study appears a little preliminary in parts and could be strengthened by additional, and entirely feasible, experiments, particularly given the potential for this study to represent a foundational piece on how regulation of oligodendrocytes might influence reward/ drug seeking behaviours.

1. As is common in the field, the study uses the term "myelin plasticity" in the title, but the investigation of myelin per se is a bit superficial. There is a reasonably subtle change to myelin as assessed by quantifying myelinated axon number and myelin thickness in response to dopaminergic neuron activation, but no data at all on myelin when it comes to the effect of morphine, for example. Given the various tools now available in the field to visualise myelin at high resolution, it would be informative to better understand how myelin per se was altered upon DA neuron

activation and morphine administration. Perhaps transgenic reporters of oligodendrocyte morphology could speak to the myelinating profiles of single cells, their number and length of myelin sheaths, or immunohistochemical analyses of markers such as caspr or others that might allow assessment of myelin sheath lengths, even together with markers of neuronal/ axonal types. At the absolute minimum, we need to understand whether the myelin changes already reported for DA neuron stimulation are seen upon morphine administration to be able to compare and contrast. I also do wonder whether this study, and the field at large, should speak to oligodendrocyte plasticity, rather than myelin plasticity, when this is principally what is being examined and reported on, and also what the key transgenic manipulation changes, i.e. oligodendrogenesis.

2. The *Myrf* mutant is not a very specific manipulation. Although it has become standard in the field, and is a powerful tool, the truth is that the conditional mutant mouse employed here as the sole regulator of "myelin plasticity" is quite a general manipulation. Upon induction the mutant transgene prevents OPCs in the adult from generating new oligodendrocytes, and this is true throughout the entire CNS, where oligodendrogenesis continues to some degree or another. Also, the manipulation is such that the cells that fail to differentiate as oligodendrocytes die, and this may create secondary consequences that could influence circuit function. Therefore, it would be very useful to have an additional manipulation that consolidated the key premise of the manuscript that activity-driven oligodendrocyte lineage regulation causally influenced reward seeking behaviour. An ideal would be to somehow influence only the responsiveness of oligodendrocytes in the VTA to Activity, but I suspect that this would require very complex experimental set up and validation. However, the corresponding author's group have previously, and very nicely, employed other methods to interrogate activity-regulated "myelination." Could their tools such as oligodendrocyte-specific TrkB receptor mutants be used, or their activity-driven BDNF mutants, or other pathway manipulations to hand. It may be that the molecular mechanism driving local VTA-based changes are difficult to disentangle at present. If so, maybe the simplest manipulation would be a gain of function manipulation, where oligodendrogenesis was stimulated, where one could test for increases in the reward seeking behaviours compared to controls. Again, various options abound, with some such as chemical treatments using clemastine now standard in the field. Whatever approach the authors choose to take, I would think it essential to have another experimental methods (either loss or gain of function, and ideally tied to activity) to corroborate the key premise of the manuscript.

Referee #2 (Remarks to the Author):

In this manuscript, the authors demonstrate that optogenetic activation of DAT positive neurons in the VTA can lead to changes in OPC proliferation and myelination, in the VTA. This appeared selective in the VTA as stimulation in the NAc did not induce any such changes. The authors then found that morphine and cocaine could drive similar changes in OPC function, and that this tracked with CPP. Finally, a genetic lesion blocking oligodendrogenesis abolished morphine CPP, which appeared to track with changes in DA release in the NAc.

This is a conceptually interesting advancement, implicating non neuronal cells in learning. The methods are strong and controls are in place. I think there are a few conceptual concerns that need

to be addressed to strengthen the authors claims.

1) The effect on morphine reward is interesting, but the authors should evaluate if 'natural' rewards such as food and social interaction induced CPP can drive similar changes in circuitry. As it reads now, it is unclear.

2) In the same vein, recent work has found that dopamine signaling is important for both morphine CPP and CPA (in Dorsal Raphe Dopamine neurons, but still). The authors should determine if it is 'reward' learning, or any drug induced learning that can drive this reward. Given that non-contingent drug can drive changes, this is an important question.

3) It is interesting that DA and dynorphin/KOR signaling do not drive changes in OPC. I was somewhat surprised that there was no evaluation of glutamate signaling as DA neurons can also co release glutamate, and glutamate has been shown to influence oligodendrogenesis. Selective deletion of glutamate release from DA neurons would be one way to examine this.

4) While the focus is on DA neurons, morphine has been thought to act via GABA neuron inhibition in the VTA, at least in part. It would close the loop to see if inhibition of GABA neurons in the VTA could drive similar changes in function.

Author Rebuttals to Initial Comments:

Response to Referees

We were delighted to see the Referees' positive comments and enthusiasm for the manuscript and are grateful for the careful review, helpful suggestions and insightful questions. We have worked to address the Referees' comments and suggestions in full, which we feel have improved and strengthened the manuscript. Below we will present in detail the major changes and then will respond to the Referee comments point-by-point.

- We found that optogenetic inhibition of GABAergic neurons in VTA, which increases dopaminergic neuron activity, promotes OPC proliferation in VTA - concordant with the effects of directly stimulating dopaminergic neuronal activity.
- To further visualize oligodendroglial response to morphine in VTA relevant to myelination, we utilized a genetic labeling strategy to sparsely label oligodendrocytes with membrane-tethered mGFP and combined this with immunohistochemistry for dopaminergic axons. Using this strategy, we can better visualize oligodendrocyte processes in relationship to dopaminergic axons. We found that morphine increases oligodendrocyte processes juxtaposed with dopaminergic axons in VTA, and increases the area of oligodendrocyte processes in VTA. These findings further support the complimentary immuno-electron microscopy data in the manuscript, indicating increased myelination of dopaminergic axons following increased dopaminergic neuronal activity.
- We blocked oligodendrogenesis using a second genetically engineered mouse model strategy, to further test the role of oligodendrogenesis in morphine-induced reward learning: We found that conditional, inducible TrkB loss from OPCs blocks oligodendrogenesis and abrogates morphine reward learning, similar to the results in the *Myrf^{-/-}* model of inducible loss of oligodendrogenesis. These findings also highlight neurotrophin signaling as a key regulator of dopaminergic neuron-OPC communication.
- In contrast to drug (morphine or cocaine)-induced reward learning, we found that food-induced reward learning does not promote OPC proliferation in VTA, and blocking oligodendrogenesis does not affect food-induced reward learning. Similarly, aversion learning using the opioid receptor antagonist naloxone does not promote OPC proliferation in VTA, and blocking oligodendrogenesis does not affect naloxone-induced conditional place aversion.
- Single nucleus RNA sequencing (sn-RNAseq) in VTA reveals robust interactions between dopaminergic neurons and OPCs. Morphine causes a transcriptional shift in oligodendroglial cells towards a more differentiated oligodendroglial cell state, consistent with morphine-induced oligodendrogenesis.
- sn-RNAseq also reveals that VTA OPCs express neither dopamine receptors nor opioid receptors, further supporting that the oligodendroglial response to dopaminergic neuronal activity or to morphine is not mediated by direct effects of DA or morphine on OPCs.

These new findings, together with additional control and verification experiments, are detailed in the point-by-point responses below.

Referee #1

"Myelin plasticity in ventral tegmental area is required for opioid reward" presents data that activation of the Dopaminergic neuron reward circuit stimulates the oligodendrocyte lineage and that this plays a role in regulating opioid-driven reward-seeking behaviour. This is a very interesting fundamental finding, and is another piece of evidence that places active regulation of the oligodendrocyte lineage as a critical aspect of neural circuit function, and in this case, arguably dysfunction. The study takes an ontogenetic-based approach in rodents to stimulate dopaminergic neuron activity in the VTA region of the brainstem and finds, quite convincingly that oligodendrocyte progenitor cells (OPCs) are stimulated to proliferate, mirroring similar observations by the same group and others in different brain areas. The study also shows that this effect is quite local, regulating OPCs in the VTA region, but not in target brain regions, e.g the NAc. Longer term ontogenetic stimulation results in local (VTA) increased oligodendrogenesis, and myelination, even though the effect sizes are somewhat muted. Very interestingly, application of either morphine or cocaine mirrors (where directly compared, see below) the effects of dopaminergic neuron stimulation on VTA oligodendrocyte lineage behaviour, suggesting the possibility that activity-driven regulation of OPCs and oligodendrocytes might play a functional role in regulating the rear circuit. This premise is supported by evidence gathered using a transgenic mouse in which the generation of new adult oligodendrocytes is impaired, such that the morphine induced changes in behaviour are muted in the mutant animal. As a first step towards investigating the potential circuit-based mechanism by which the oligodendrocyte lineage may influence behaviour, the study also shows that dopamine release in the NAc is reduced in the transgenic animals with impaired oligodendrogenesis. To me, this study has the potential to be of enormous general interest. The observations are novel, the topic of importance and the data as presented so far appear robust, although I will note a lack of expertise in rodent behavioural analyses that mute my own ability to speak to the details thereof. Nonetheless, the study appears a little preliminary in parts and could be strengthened by additional, and entirely feasible, experiments, particularly given the potential for this study to represent a foundational piece on how regulation of oligodendrocytes might influence reward/ drug seeking behaviours.

1. As is common in the field, the study uses the term "myelin plasticity" in the title, but the investigation of myelin per se is a bit superficial. There is a reasonably subtle change to myelin as assessed by quantifying myelinated axon number and myelin thickness in response to dopaminergic neuron activation, but no data at all on myelin when it comes to the effect of morphine, for example. Given the various tools now available in the field to visualise myelin at high resolution, it would be informative to better understand how myelin per se was altered upon DA neuron activation and morphine administration. Perhaps transgenic reporters of oligodendrocyte morphology could speak to the myelinating profiles of single cells, their number and length of myelin sheaths, or immunohistochemical analyses of markers such as caspr or others that might allow assessment of myelin sheath lengths, even together with markers of neuronal/ axonal types. At the absolute minimum, we need to understand whether the myelin changes already reported for DA neuron stimulation are seen upon morphine administration to be able to compare and contrast. I also do wonder whether this study, and the field at large, should speak to oligodendrocyte plasticity, rather than myelin plasticity, when this is principally what is being examined and reported on, and also what the key transgenic manipulation changes, i.e. oligodendrogenesis.

We thank the reviewer for this comment. We have both expanded our studies and referred to oligodendroglial changes in the text where appropriate. In our manuscript we included MBP staining

quantification in ventral tegmental area (VTA) which indicates a general change in myelin content (Manuscript Fig 2f-g). As the confocal micrograph on Manuscript Fig 2f shows, it is difficult to analyze myelin internodes using immunohistochemistry in mouse tissue, especially in an area like VTA which lacks bundles of axons running in the same direction. Dopaminergic axons in the ventral tegmental area lie both in the anterior-posterior plane and medial-lateral plane, such that axons travel in different directions, which complicates internode analysis. To better understand the myelin changes, we performed immunoEM microscopy analysis (Manuscript Fig. 2h-j) which demonstrated that medium sized dopaminergic neurons are becoming myelinated as a result of increased dopaminergic neuron activity.

To address these important questions regarding myelin sheath dynamics in morphine-evoked oligodendrogenesis, we adopted a genetic fluorescence labeling strategy (Rev. Fig. 1). We crossed mTdtomato-mGFP^{flx/flx} animals with Plp1Cre^{ERT} and conditionally induced expression of membrane-tethered mGFP at 4 weeks of age to label differentiating oligodendrocytes. 3 weeks after tamoxifen induction, we performed conditional place preference with saline and morphine (Rev. Fig. 1a). As expected, morphine increased locomotor sensitivity and morphine conditioned animals acquired a preference for the morphine-paired chamber (Rev. Fig. 1b-d). We found that mGFP-labelled oligodendrocytes are juxtaposed to/appear to be interacting with TH⁺ dopaminergic neuron axons in VTA and appear to be wrapping around proximal parts of dopaminergic axons (Rev. Fig. 1e). Each oligodendrocyte in this region interacts with multiple dopaminergic axons and appear ramified. Morphine increased the number of mGFP⁺ oligodendrocytes wrapping around TH⁺ axons (Rev. Fig. 1f, g) and morphine increased the quantity (area) of mGFP⁺ oligodendrocyte processes in VTA (Rev. Fig. 1h).

In addition to this transgenic strategy, we also tried a labeling strategy that was previously used in the field (Osso et al., 2021): map1^{flx/flx} crossed with PdgfraCre^{ERT}. However, we did not see any baseline labelled oligodendrocytes in VTA either 5 days or 3 weeks after tamoxifen induction of expression at 4 weeks or 7 weeks of age, so this strategy was not further pursued.

Review Figure 1. Morphine increases oligodendroglial processes around dopaminergic axons in VTA. **a**, Schematic of experimental paradigm to fluorescently label oligodendrocytes ($mTtdtomato-mGFP^{flx/flx}; Plp1Cre^{ERT}$) followed by morphine conditioning. **b**, Morphine increases locomotor sensitization. **c**, Morphine treated mice acquire a strong preference to morphine-paired chamber whereas saline treated mice do not. **d**, Morphine-conditioned mice show increased CPP score compared to saline treated mice. **e**, Confocal micrographs of morphine treatment depicting TH labeled dopaminergic neurons (grey) and mGFP labeled oligodendrocytes (green) in VTA. A dopaminergic neuron with a mGFP⁺ oligodendrocyte juxtaposed to the proximal portion of its axon in VTA. **f**, Confocal micrographs depicting TH labeled dopaminergic neurons (grey) and mGFP labeled oligodendrocytes (green) in VTA. **g**, Morphine conditioned mice show increased mGFP⁺ oligodendrocytes that are juxtaposed to/interacting with TH⁺ axons in VTA. **h**, Morphine-conditioned mice show increased area of mGFP⁺ oligodendrocyte processes in VTA. scale bar = 50 μ m.

2. The Myrf mutant is not a very specific manipulation. Although it has become standard in the field, and is a powerful tool, the truth is that the conditional mutant mouse employed here as the sole regulator of "myelin plasticity" is quite a general manipulation. Upon induction the mutant transgene prevents OPCs in the adult from generating new oligodendrocytes, and this is true throughout the entire CNS, where oligodendrogenesis continues to some degree or another. Also, the manipulation is such that the cells that fail to differentiate as oligodendrocytes die, and this may create secondary consequences that could influence circuit function. Therefore, it would be very useful to have an additional manipulation that consolidated the key premise of the manuscript that activity-driven oligodendrocyte lineage regulation causally influenced reward seeking behaviour. An ideal would be to somehow influence only the responsiveness of oligodendrocytes in the VTA to Activity, but I suspect that this would require very complex experimental set up and validation. However, the corresponding

author's group have previously, and very nicely, employed other methods to interrogate activity-regulated "myelination." Could their tools such as oligodendrocyte-specific TrkB receptor mutants be used, or their activity-driven BDNF mutants, or other pathway manipulations to hand. It may be that the molecular mechanism driving local VTA-based changes are difficult to disentangle at present. If so, maybe the simplest manipulation would be a gain of function manipulation, where oligodendrogenesis was stimulated, where one could test for increases in the reward seeking behaviours compared to controls. Again, various options abound, with some such as chemical treatments using clemantine now standard in the field. Whatever approach the authors choose to take, I would think it essential to have another experimental methods (either loss or gain of function, and ideally tied to activity) to corroborate the key premise of the manuscript.

These are excellent points. To address these comments, we did the following: 1. We did attempt a local knockdown of Myrf using an shRNA strategy, but encountered technical difficulties transducing oligodendroglia. 2. We also tested clemastine in addition to morphine to augment oligodendrogligenesis, but this did not further improve conditioned place preference (The CPP behavioral test is already "maxed out" by 10mg/kg morphine exposure. The effect of morphine is so strong in this assay that even doubling the dose of morphine – which does further increase oligodendrogenesis - does not further increase the conditioned place preference.) 3. We tested an alternative strategy to block activity-regulated oligodendrogenesis and myelination using the OPC-specific *TrkB* knock out mouse model, as suggested. Encouragingly, this experiment replicated the findings in the Myrf-KO model; with OPC-specific deletion of *TrkB* to block the activity-regulated response, mice did not exhibit conditioned place preference with morphine administration. We appreciate these helpful suggestions, and feel that these complimentary data using an alternative model to block oligodendrogenesis adds an important validation to the study. Below, all 3 experiments are summarized.

1. First, to eliminate the global effects of blocking oligodendrogenesis, we initially attempted to knockdown Myrf specifically in VTA and used lentiviral shRNA strategy (Sigma, Myrf MISSION shRNA Lentivirus, Myrf-CMV-tGFP, viral titer:10⁹). We injected 1 μ l of lentivirus into VTA and analyzed the brains 3 days or 7 days later. Even though there were transduction of other cell types in the region, unfortunately, we did not observe an efficient transduction of oligodendroglial cells (Rev. Fig. 2).

Review Figure 2. shRNA strategy for Myrf knockdown in oligodendroglial cells is not efficient in VTA. a, Confocal micrograph of VTA that is injected with shRNA-Myrf lentivirus particles. Green cells expressing GFP (green) indicating a successful transduction (yellow arrowheads), OPCs (grey), oligodendroglial cells (red). scale bar = 50 μ m. Note the lack of GFP expression in oligodendroglial lineage (Olig2+) cells.

2. In response to the reviewer's excellent suggestion, we employed a gain-of-function strategy using Clemastine injections to increase oligodendrogenesis. We injected animals once daily for 3 weeks with clemastine (10 mg/kg) (Pan et al., 2020) or saline as a control and performed morphine conditioning as we continued to inject animals once daily with clemastine until the end of CPP test. Both saline and clemastine treated animals showed strong preference towards the conditioning chamber (Rev. Fig. 3a-c). However, quantification of a stronger learning with this behavioral test is not straightforward to quantify in preference analysis. This is also evident in our previous data where we administered different concentrations of morphine (10 mg/kg and 20 mg/kg) (Manuscript Fig. 3d-g) where we do not see a quantifiable difference in CPP Scores even though we do see increased locomotion effect in higher morphine doses. Here, we did not observe a significant difference in CPP score between saline and clemastine injected groups to indicate a gain-of-function effect in the reward learning.

Review Figure 3. Clemastine administration to increase oligodendrogenesis does not further increase morphine-induced reward learning. **a**, Schematic of experimental paradigm for daily clemastine treatment and concurrent morphine conditioning. **b**, Both saline and clemastine treated animals acquire a strong preference to morphine-paired chamber. **c**, Clemastine treated animals show similar levels of CPP score to saline treated animals after morphine conditioning.

3. As suggested, we used another animal model to conditionally block oligodendrogenesis. We previously demonstrated that TrkB-Bdnf signaling is critical for activity-regulated myelination in premotor cortex (Geraghty et al., 2019). As the reviewer also suggested, we used conditional knockout of TrkB receptors from OPCs ($TrkB^{flx/flx}; Pdgfra^{Cre^{ERT1}}$) to block oligodendrogenesis immediately prior to morphine CPP (Rev. Fig. 4a). Similar to our findings with Myrf model (Manuscript Fig. 4), oligodendroglial TrkB loss did not affect locomotor sensitization (Rev. Fig. 4b), but it abrogated reward learning (Rev. Fig. 4c, d). As OPC-TrkB KO results in impaired attention and memory behavior in the novel object recognition test (NORT) by 4 weeks after inducing TrkB loss in OPCs (Geraghty et al., 2019), we confirmed that attention and memory is intact at one week after TrkB loss in OPCs here (Rev. Fig. 4e, f) Consistent with the effect on CPP, TrkB loss also attenuated the proliferative response of OPCs ($EdU^+ Pdgfra^+$) to morphine in VTA (Rev. Fig. 4g, h). These results strengthen our findings that oligodendrogenesis is necessary for morphine reward and indicate a potential mechanism (BDNF-TrkB signaling) by which oligodendroglia can respond to dopaminergic neuronal activity.

Review Figure 4. Oligodendroglial TrkB loss abrogates morphine-induced reward learning. **a**, Schematic of experimental paradigm to conditionally knockout TrkB ($TrkB^{flx/flx}; PdgfraCre^{ERT}$) from OPCs immediately prior to morphine conditioning. **b**, Both control and $TrkB^{-/-}$ animals show locomotor sensitization with morphine conditioning. **c**, Control mice acquire a strong preference for the morphine conditioning chamber whereas $TrkB^{-/-}$ animals do not. **d**, $TrkB^{-/-}$ animals show decreased CPP Score compared to control littermates. **e**, Schematic of experimental paradigm for novel object recognition test; mice are introduced to two identical objects and tested 24 hours after for memory by switch one of the objects with a novel object. 1 week after the end of tamoxifen administration $TrkB^{-/-}$ ($TrkB^{-/-}; Pdgfra-Cre^{ERT}$) animals and littermate controls are tested in novel object recognition test. This timeline is identical to CPP paradigms tested above in panels a-d. **f**, Both control and $TrkB^{-/-}$ animals spend more time with novel object compared to the familiar object (control, n=12 mice; $Myrf^{-/-}$, n=11 mice). **g**, Confocal micrographs depicting OPC proliferation in VTA after morphine CPP. TH labeling dopaminergic neurons (green), Pdgfra labeling OPCs (grey), and EdU labeling proliferating cells (red). Arrowheads show proliferating OPCs. **h**, $TrkB^{-/-}$ mice show decreased new OPCs ($Edu^{+} Pdgfra^{+}$) in VTA compared to control mice. scale 50 μ m

Referee #2

In this manuscript, the authors demonstrate that optogenetic activation of DAT positive neurons in the VTA can lead to changes in OPC proliferation and myelination, in the VTA. This appeared selective in the VTA as stimulation in the NAc did not induce any such changes. The authors then found that morphine and cocaine could drive similar changes in op function, and that this tracked with CPP. Finally, a genetic lesion blocking oligodendrogenesis abolished morphine cpp, which appeared to track with changes in DA release in the Nac.

This is a conceptually interesting advancement, implicating non neuronal cells in learning. The methods are strong and controls are in place. I think there are a few conceptual concerns that need to be addressed to strengthen the authors claims.

1) The effect on morphine reward is interesting, but the authors should evaluate if 'natural' rewards such as food and social interaction induced CPP can drive similar changes in circuitry. As it reads now, it is unclear.

We thank for this important point from the reviewer. To delineate whether oligodendrogenesis is necessary only for drug-evoked learning or whether it can play a role in natural rewards, we performed food-evoked conditional preference test. First, we conditioned animals for food reward, and included both ad libitum controls and food-restricted controls to compare with reward animals which were food-restricted prior to food reward. We administered EdU during conditioning sessions to all animals to track cell division (Rev. Fig. 5a). Animals that received food reward acquired a strong preference to the conditioning chamber, while both of the control groups did not (Rev. Fig. 5b-c). Analysis of OPC proliferation in VTA showed similar levels of new OPCs (EdU⁺ Pdgfra⁺) in the food reward group compared to control groups (Rev. Fig. 5d).

Second, to better understand whether oligodendrogenesis is necessary for food-induced reward learning, we blocked oligodendrogenesis by conditionally knocking Myrf from OPCs 4 weeks prior to food-reward conditioning (Rev. Fig. 5e). Both Myrf^{-/-} animals and control littermates acquired a strong preference to the food-paired chamber (Rev. Fig. 5f-g). Collectively, these findings demonstrate that oligodendrogenesis is not required for food-induced natural reward learning and indicate a specific role for oligodendrogenesis in drug-evoked reward learning.

Review Figure 5. Oligodendrogenesis is not necessary for food-induced reward learning. **a**, Schematic of experimental paradigm for food reward. **b**, Food reward conditioned animals show a strong preference to food-paired chamber whereas ad libitum or food restriction control animals do not. **c**, Food reward conditioned animals show increased CPP Score compared to ad libitum and food restriction controls. **d**, Food reward conditioned animals show lower levels of new OPCs (EdU⁺ Pdgfra⁺) in VTA compared to food restricted controls, but similar levels to ad libitum controls. **e**, Schematic of experimental paradigm for food reward with oligodendrogenesis blocking Myrf^{flx/flx}; PdgfraCre^{ERT} animals and littermate controls. **f**, Both control and Myrf^{-/-} animals acquire a strong preference to the food reward-paired conditioning chamber. **g**, CPP Scores for food reward conditioning are similar in control and Myrf^{-/-} animals. **h**, Control and Myrf^{-/-} animals show similar levels of new OPCs (EdU⁺ Pdgfra⁺) in VTA after food reward conditioning.

This was a very interesting finding that extends and strengthens the manuscript, and we are grateful to the reviewer for the excellent suggestion. Why it is that drug-induced reward (or optogenetic stimulation of all VTA dopaminergic neurons) evokes an oligodendroglial response, while food reward does not will be a topic for future study, and we have added discussion about this point. It may be that different subpopulations of dopaminergic neurons are activated by drug reward and food reward, just as different subpopulations of dopaminergic neurons projecting to different regions of NAc mediate conditioned place preference and conditioned place aversion (Garritsen et al., 2023, *Nat. Revs. Neuro.*). Alternatively, or in addition, it could be that the strength of the stimulus determines the oligodendroglial response, with optogenetic stimulation and morphine or cocaine administration evoking a greater DA neuronal response than chocolate.

2) In the same vein, recent work has found that dopamine signaling is important for both morphine CPP and CPA (in Dorsal Raphe Dopamine neurons, but still). The authors should determine if it is 'reward' learning, or any drug induced learning that can drive this reward. Given that non-contingent drug can drive changes, this is an important question.

To address this question, we used kappa opioid receptor antagonist (naloxone)-induced conditioned place aversion (CPA) (Rev. Fig. 6a). First, we conditioned animals with naloxone, or saline as control, and administered EdU during conditioning sessions to trace dividing cells. Naloxone-treated animals acquired a strong avoidance towards the naloxone-paired chamber, while saline-treated animals did not show any preference (Rev. Fig. b-c). Naloxone conditioning did not change the levels of new OPCs (EdU⁺ Pdgfra⁺) in VTA, suggesting that naloxone does not cause an oligodendroglial response.

Second, we tested whether blocking oligodendrogenesis affects naloxone-induced conditioned place aversion and utilized the Myrf model to conditionally block oligodendrogenesis 4 weeks prior to naloxone conditioning experiments (Rev. Fig. 6e). Both Myrf^{-/-} animals and control littermates acquired a strong aversion to naloxone-paired conditioning chamber (Rev. Fig. 6f, g) indicating that blocking oligodendrogenesis does not affect aversion learning. Consistently, both control and Myrf^{-/-} animals showed similar levels of new OPCs (EdU⁺ Pdgfra⁺) in VTA (Rev. Fig. 6h).

Review Figure 6. Oligodendrogenesis is not necessary for naloxone-evoked aversion. **a**, Schematic of experimental paradigm for kappa opioid receptor antagonist, naloxone conditioning. **b**, Naloxone administered animals show a strong avoidance to naloxone-paired chamber whereas saline administered control animals do not. **c**, Naloxone conditioned animals show decreased CPP Score compared to saline controls. **d**, Saline and naloxone administered animals show similar levels of new OPCs (Edu⁺ Pdgfra⁺) in VTA. **e**, Schematic of experimental paradigm for naloxone conditioning with oligodendrogenesis blocking Myrf^{flx/flx}; PdgfraCre^{ERT} animals and littermate controls. **f**, Both control and Myrf^{-/-} animals acquire a strong avoidance to the naloxone-paired conditioning chamber. **g**, CPP Scores for naloxone conditioning are similar in control and Myrf^{-/-} animals. **h**, Control and Myrf^{-/-} animals show similar levels of new OPCs (Edu⁺ Pdgfra⁺) in VTA after naloxone conditioning.

3) It is interesting that DA and dynorphin/KOR signaling do not drive changes in OPC. I was somewhat surprised that there was no evaluation of glutamate signaling as DA neurons can also co release glutamate, and glutamate has been shown to influence oligodendrogenesis. Selective deletion of glutamate release from DA neurons would be one way to examine this.

This raises the important point of determining the mechanisms by which oligodendroglia respond to morphine and VTA neuronal activity changes. The communication routes between oligodendroglia and neurons are likely to be multifaceted and heterogenous across different brain regions. We previously identified Bdnf-TrkB signaling between cortical projection neurons and OPCs as a critical mechanistic component of activity-regulated myelination (Geraghty et al., 2019). We have now tested this mechanism in VTA, and now demonstrate that TrkB signaling in OPCs is also required for morphine-induced oligodendrogenesis and morphine reward learning (Rev. Fig. 4g-l).

To further explore mechanisms communication between VTA neurons and oligodendroglia, we performed single nucleus RNA sequencing (snRNAseq) of VTA after morphine CPP (Rev. Fig. 7a). We identified the expected cell types of VTA including dopaminergic neurons, glutamatergic neurons, GABAergic neurons, OPCs, oligodendrocytes, astrocytes (Rev. Fig. 7b). In this data set, we also identified different OPC and oligodendrocyte cellular subpopulations consistent with previous single cell studies (Marques et al., 2016) (Rev. Fig. 7c). Using CellChat computational analysis (Jin et al, 2021) we determined cell-to-cell interactions between cell types in VTA. This analysis revealed a high level of interactions between dopaminergic neurons and OPCs (Rev. Fig. 7d) and suggested a variety of signaling pathways that are utilized across different cell types (Rev. Fig. 7e).

We found that VTA oligodendroglia did not express any type of dopamine receptors (Drd1-5) (Rev. Fig. 8a). Additionally, VTA oligodendroglia did not express Oprk1, Oprm1, or Oprd1 in saline nor morphine treated animals (Rev. Fig. 8b), consistent with our findings demonstrating that morphine-induced oligodendrogenesis does not occur through Oprk1 and that neither dopamine nor morphine directly affects OPC proliferation *in vitro*.

snRNAseq analyses show that TrkB is highly expressed in OPCs and to a lower extent in oligodendrocytes, which is consistent with our findings that OPC-specific TrkB loss abrogates morphine reward (Rev. Fig. 9a). Meanwhile, the TrkB ligand Bdnf is mostly expressed by dopaminergic, glutamatergic and GABAergic neurons in the VTA (Rev. Fig. 9a) suggesting the possibility that blocking Bdnf-TrkB signaling by conditional deletion of TrkB from OPCs blocks the communication between neurons and OPCs which results in abrogated reward learning. To better understand the neurotransmitter-mediated responses in oligodendroglia, we analyzed the receptors of dopamine, GABA, and glutamate across oligodendroglia in both saline and morphine conditions. We did not detect any significant changes in expression levels of any of these neurotransmitter receptors in response to morphine (Rev. Fig. 9b).

Morphine caused shifts across oligodendroglial clusters accompanied by gene expression changes that are consistent with oligodendroglial differentiation. For example, with morphine exposure we found that cluster 4 is shifted towards cluster 9, which is defined by upregulation of Wnt signaling and lamellipodium assembly and organization that are in line with oligodendroglial differentiation (Rev. Fig. 10a, b, c). Similarly, cluster 6 cells shifted towards cluster 11 (Rev. Fig. 10d, e, f), and cluster 13 cells shifted towards 7 (Rev. Fig. 10g, h, i), both of which are changes suggesting a transcriptional program consistent with more differentiated oligodendroglial clusters.

Overall, our snRNAseq data suggest that neuron-oligodendroglial interactions are complex and there is robust communication between dopaminergic neurons and OPCs. These snSeq data are consistent with our functional data showing that neurotrophin (TrkB) signaling is one of the mechanisms that mediates morphine-induced oligodendrogenesis in VTA.

Review Figure 7. snRNAseq identifies VTA cell types and interactions amongst them. a, Schematic of experimental paradigm for snRNAseq after morphine CPP. **b,** snRNAseq identifies a variety of cell populations in VTA. **c,** snRNAseq identifies oligodendroglial subpopulations which indicates different stages of differentiation and maturity. **d,** CellChat interaction maps showing the interactions of dopaminergic neurons and OPCs. Thickness of the lines represents number of signaling going from each cell type to others. Size of the circle representing cell populations are in proportion to the size of the population. **e,** Relative pathways graph (CellChat) suggest the most utilized signaling pathways amongst cell populations and indicates a proportion of each signaling for saline and morphine groups. Gene names in salmon text are upregulated in the saline group, gene

names in teal text are upregulated in the morphine group, and gene names in black are not significantly different between groups; paired Wilcoxon test with significance cut off of $p=0.05$.

Review Figure 8. Oligodendroglial cells in VTA do not express dopamine receptors or opioid receptors. a-b, UMAP of single nucleus sequencing (sn-Seq) in VTA. Cell cluster identity is labeled in the first panel. a, Dopamine receptors (Drd1-5) are not expressed by oligodendrocytes or OPCs in the VTA, and this is not affected by morphine treatment. Drd2 is highly expressed by dopaminergic

neurons. **b**, Opioid receptors (*Oprk1*, *Oprm1*, *Oprd1*) are not expressed by oligodendrocytes or OPCs in the VTA, and this is not affected by morphine treatment. *Oprk1*, and *Oprm1* are highly expressed by different neuronal populations in the region.

Review Figure 9. Oligodendroglial cells in VTA express *TrkB*. **a**, left: UMAP of oligodendroglia in VTA. snRNAseq reveals that OPCs in VTA express high levels of *TrkB*, and oligodendrocytes also express *TrkB* but to a lesser extent. Right: UMAP of all cell types in VTA. *Bdnf* is expressed mainly by neurons including dopaminergic neurons, GABAergic neurons, and glutamatergic neurons. **b**, Heatmap of neurotransmitter receptor gene expression in VTA oligodendroglia by snRNA-seq.

Oligodendroglial cells express different levels of glutamatergic, and GABAergic receptors, but not dopaminergic receptors. This expression pattern is not affected by morphine treatment.

Review Figure 10. Morphine causes transcriptional shifts in oligodendroglial gene expression that is consistent with differentiation. **a**, snRNAseq reveals that morphine causes cluster 4 oligodendrocytes to shift towards cluster 9. **b**, Differentially expressed genes in cluster 4 vs 9. **c**, Morphine causes upregulation of lamellipodium-related genes, and Wnt signaling and downregulation of transmembrane transport-related genes. **d**, Morphine causes cluster 6 oligodendrocytes to shift towards cluster 11. **e**, Differentially expressed genes in cluster 6 vs 11. **f**, Morphine causes upregulation of biosynthetic and metabolic process related genes, and downregulation of synapse assembly genes. **g**, Morphine causes cluster 13 oligodendrocytes to shift towards cluster 7. **h**, Differentially expressed genes in cluster 13 vs 7. **i**, Morphine causes upregulation of exocytosis related genes, and downregulation of synaptic signaling related genes.

4) While the focus is on DA neurons, morphine has been thought to act via GABA neuron inhibition in the VTA, at least in part. It would close the loop to see if inhibition of GABA neurons in the VTA could drive similar changes in function.

To address this excellent question, we optogenetically inhibited GABAergic neurons in VTA. We injected Cre inducible NpHR::YFP (AAV-Ef1a-DIO-NpHR::YFP) or YFP control (AAV-Ef1a-DIO-YFP) into the VTA of *Vgat-Cre* mice, and administered EdU to track cell division immediately prior to optogenetic stimulations. This strategy achieves GABAergic neuron-specific expression of NpHR for optogenetic inhibition (Rev. Fig. 11a; Manuscript Extended Fig. 2). Optogenetic inhibition of GABAergic neurons for 30 minutes increased the density of new oligodendroglia ($\text{EdU}^+ \text{Olig2}^+$) and new OPCs ($\text{EdU}^+ \text{Pdgfra}^+$) in the VTA after 3 hours (Rev. Fig. 11b, c; Manuscript Extended Data Fig. 2). These findings demonstrate that inhibiting the GABAergic inhibition on dopaminergic neurons in VTA also results in an increase in new OPCs.

Review Figure 11. Inhibition of GABAergic neurons in VTA increases OPC proliferation. **a**, Schematic of experimental paradigm for inhibiting GABAergic neurons in VTA; 30-min optogenetic inhibition session followed by brain analysis 3 hours after the end of optogenetic paradigm. **b**, Inhibition of GABAergic neurons increases new oligodendroglia ($\text{EdU}^+ \text{Olig2}^+$) and **c**, newly proliferated OPC density ($\text{EdU}^+ \text{Pdgfra}^+$) in VTA. * $p < 0.05$. Data shown as mean, error bars indicate SEM.

Reviewer Reports on the First Revision:

Referees' comments:

Referee #1 (Remarks to the Author):

This is an exceptionally strong revision that now includes a series of additional experimental approaches that address reviewer comments. From my own perspective, the finding that conditional targeting of TrkB signalling in oligodendrocytes diminishes morphine-induced reward is a hugely reassuring validation of the single initial experimental approach pointing to that conclusion documented in the original submission.

In my opinion, the manuscript is now suitable for publication, and I think it represents a hugely important finding of general interest.

A minor lingering point, that really is just semantics, and for the authors to take a final view on: I do wonder still if speaking about plasticity of oligodendrocytes rather than myelin plasticity might be a bit more accurate from a high level point of view. Oligodendrocyte plasticity includes myelin plasticity, and so nothing is lost by making such a change, and it could be that the mechanisms by which oligodendrocytes ultimately influence the relevant reward circuitry reflect functions not specific to the plasticity of myelin sheathes per se, but other roles of oligodendrocytes. Something to consider in the context of title and abstract perhaps.

Referee #2 (Remarks to the Author):

The authors have done a very nice job of addressing my concerns using new experimental data. This is an important addition to the literature.